# Uncertainty-Sensitive Privileged Learning

**Fan-Ming Luo    Lei Yuan    Yang Yu**[*]

National Key Laboratory for Novel Software Technology, Nanjing University, China
School of Artificial Intelligence, Nanjing University, China
Polixir.ai
luofm@lamda.nju.edu.cn, yuanl@lamda.nju.edu.cn, yuy@nju.edu.cn

## Abstract

Privileged learning efficiently tackles high-dimensional, partially observable decision-making problems by first training a privileged policy (PP) on low-dimensional privileged observations, and then deriving a deployment policy (DP) either by imitating the PP or coupling it with an observation encoder. However, since the DP relies on local and partial observations, a behavioral divergence (BD) often emerges between the DP and the PP, ultimately degrading deployment performance. A promising strategy is to train a PP to learn the optimal behaviors attainable under the DP's observation space by applying reward penalties in regions with large BD. However, producing these behaviors is challenging for the PP because they rely on the DP's information-gathering progress, which is invisible to the PP. In this paper, we quantify the DP's information-gathering progress by estimating the prediction uncertainty of privileged observations reconstructed from partial observations, and accordingly propose the framework of Uncertainty-Sensitive Privileged Learning (USPL). USPL feeds this uncertainty estimation to the PP and combines reward transformation with privileged-observation blurring, driving the PP to choose actions that actively reduce uncertainty and thus gather the necessary information. Experiments across nine tasks demonstrate that USPL significantly reduces the behavioral discrepancies, achieving superior deployment performance compared to baselines. Additional visualization results show that the DP accurately quantifies its uncertainty, and the PP effectively adapts to uncertainty variations. Code is available at https://github.com/FanmingL/USPL.

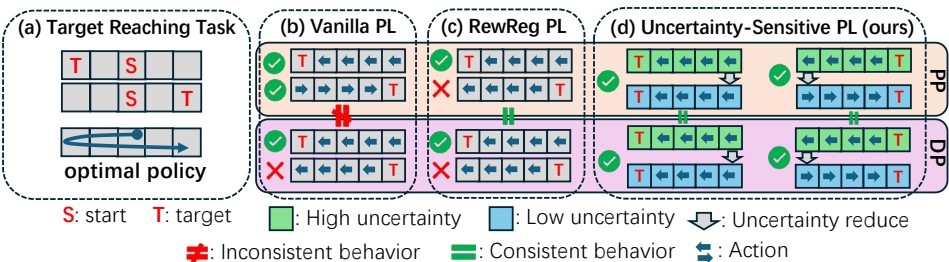

Figure 1: Policy comparison among different privileged learning (PL) methods on the Target Reaching task. The privileged policy (PP) observes both agent and target positions, whereas the deployment policy (DP) sees only the agent's position. (a) Task setup: the target randomly appears at one of two locations, with a slight leftward bias. (b–d) PP and DP behaviors under Vanilla PL (b), Reward-Regularized PL (c), and our Uncertainty-Sensitive PL (d); in (d), uncertainty falls as the agent approaches the boundary, and the policies adapt accordingly.

---

[*]Corresponding author.

39th Conference on Neural Information Processing Systems (NeurIPS 2025).

# 1 Introduction

Real-world decision-making is typically partially observable and often relies on high-dimensional visual observations, such as in autonomous driving [1], quadruped locomotion [2, 3, 4], and household robotics [5], all of which pose significant challenges to reinforcement learning (RL) [6], introducing substantial instability and inefficiency [7, 8, 9]. The privileged learning framework [10] mitigates these challenges by leveraging low-dimensional privileged observations to either complement missing state information [11, 12, 13, 14] or replace high-dimensional observations [15], thereby enabling efficient training of a privileged policy (PP). The resulting PP is then leveraged to construct a deployment policy (DP), either by training the DP to imitate the PP [16] or by combining the PP with an observation encoder that predicts privileged observations [13]. Since the DP does not have access to privileged observations, the asymmetric observation spaces between PP and DP often result in behavioral discrepancies that degrade deployment performance [16]. As illustrated in Fig. 1, in a Target Reaching task, the agent is required to navigate to a hidden target location based solely on partial observations. The PP has access to the full target position, while the DP only observes the agent's location. In Fig. 1(b), with the vanilla method, the PP can learn to successfully reach the target under both target position settings based on privileged observations. However, the behavior of the PP cannot be replicated by the DP, as the DP lacks information about the target's location. Simply imitating the PP's behavior results in suboptimal performance.

To reduce the behavioral divergence (BD) between PP and DP, two main approaches have been proposed. The first improves DP using RL—either via RL-based fine-tuning [15] or joint optimization of imitation and RL objectives [17, 16, 18, 19]. While effective in low-dimensional or mildly asymmetric cases, these methods struggle in high-dimensional tasks with large BD due to RL instability, leading to inefficient training. The second approach targets the PP, making it easier to imitate [20, 2, 21]. Directly training a PP to produce the best behavior achievable within the DP's observation space (the optimal behavior that can be replicated by the DP) via reward regularization [20] is a promising and scalable direction, as it avoids exploration and RL on the DP. However, the optimal behavior of the DP may depend on its progress in gathering privileged information. For example, the optimal policy in Fig. 1(a) initially moves to the left until it reaches the left boundary, and if the episode has not ended, indicating that the target is on the right, it then turns back. Such behavior is actually reliant on its progress in gathering privileged information. When it is impossible to infer privileged observations based on its historical observations, the DP executes a fixed exploration behavior (moving left). Once it reaches the left boundary, the DP can infer that the target must be on the right, and it will then move towards the target. However, the PP cannot know the progress of gathering privileged information through its instantaneous observations, making it challenging for the PP to learn the optimal behavior of the DP. As a result, although Reward Regularization PL (Fig. 1(c)) maintains behavioral consistency between the DP and the PP, it still fails to achieve both objectives. Using memory-based architectures such as RNNs [22, 23] or Transformers [24] may help, but these methods suffer from inefficient training and require careful hyperparameter tuning [25, 26].

In this paper, we propose the framework of Uncertainty-Sensitive Privileged Learning (USPL). It employs an observation encoder to reconstruct privileged observations from the DP's partial observations and then estimates the uncertainty of these reconstructions, which quantifies how much information still needs to be gathered. USPL augments PP's observation with the estimated uncertainty, and further leverages reward transformation and privileged blurring to incentivize the PP to produce behaviors that actively gather information by minimizing uncertainty. These techniques encourage the PP to learn DP's optimal behavior based on the uncertainty estimation. We empirically validate USPL across nine diverse tasks, including quadruped locomotion, quadrotor hovering, and robotic manipulation with both state-based and visual inputs. Our results show that USPL significantly reduces behavioral discrepancies between the PP and DP (as in Fig. 1(d)), yielding higher deployment success rates than baseline methods. Furthermore, we observe that the DP accurately estimates its uncertainty, and the PP rapidly adapts its behavior in response to changes in uncertainty.

# 2 Background and Related Work

## 2.1 Partially Observable Markov Decision Processes

Partially Observable Markov Decision Processes (POMDPs) [27, 28, 29] provide a principled framework for decision-making under uncertainty and incomplete state information. A POMDP is defined

by the tuple $\langle \mathcal{S}, \mathcal{A}, \mathcal{O}, P, O, r, \gamma, \rho_0 \rangle$, where $\mathcal{S}$, $\mathcal{A}$, and $\mathcal{O}$ denote the state, action, and observation spaces, $P$ and $O$ are the transition and observation functions, $r$ is the reward function, $\gamma$ is the discount factor, and $\rho_0$ is the initial-state distribution. Because the agent observes only $o \in \mathcal{O}$, it must reason over a belief state—a distribution over $\mathcal{S}$—to maximize expected cumulative return.

Mainstream RL algorithms for POMDP approximate the belief state using histories of observations and actions, typically with memory-augmented neural architectures such as RNNs [30, 22] and Transformers [24]. These methods work well on low-dimensional tasks with modest memory requirements [27, 31, 32, 33, 34, 35]; however, training becomes increasingly inefficient and unstable as task complexity grows [25, 32]. Luo et al. [25] showed that RNN-based RL demands finely tuned learning rates for stability, while Parisotto et al. [26] reported that Transformer-based RL requires specialized architectural modifications. Real-world applications further complicate matters by coupling long-horizon memories with high-dimensional sensory inputs such as images [3, 36, 15].

## 2.2 Privileged Learning

Given the inefficiencies of memory-augmented RL in complex, high-dimensional POMDPs, *privileged learning* [10] has emerged as a promising alternative. By introducing *privileged observations*—compact, globally informative signals available only during training—privileged learning mitigates the sample inefficiency and instability of standard RL and enables the efficient training of a *privileged policy* (PP). During deployment, these privileged signals are no longer available, so a separate *deployment policy* (DP) must act solely on partial observations. In practice, the DP is typically obtained either by (i) directly imitating the PP through supervised learning [16], or (ii) learning an encoder that reconstructs privileged observations from partial inputs and feeding them to the PP [13]. Both strategies retain the decision-making power of RL while leveraging the stability of supervised learning, and have thus been widely adopted in real-world robotic control tasks [13, 36, 3, 2]. Nonetheless, a persistent challenge is the behavioral discrepancy between the PP and DP, which stems from the asymmetry in their observation spaces and can significantly impair deployment decision quality [16]. Two main research lines have been proposed to address this issue.

One research line fine-tunes the DP with reinforcement-learning losses [15] or blends RL and imitation objectives during training [17, 16, 18, 19]. Balancing these two losses [16, 18] and designing effective reward shaping [19] are persistent challenges. When the PP–DP gap is large, however, these methods eventually fall back on RL, inheriting its instability and exploration burden. A complementary line of work focuses on optimizing the PP so that it is easier for the DP to imitate. One effective strategy reduces the PP's reliance on privileged observations by mapping them into a latent space with a privileged encoder and then compressing that space through latent alignment [7] or an information bottleneck [21]. Alternatively, reward-penalty methods [20] discourage behaviors that the DP struggles to replicate by penalizing the PP in regions where DP errors are high, effectively searching—via RL—for behaviors that minimize behavior divergence. Because they avoid applying RL directly to the DP, these approaches are highly scalable. Nevertheless, without explicit access to the DP's progress in gathering privileged information, aligning the PP with the DP's optimal behavior remains challenging, which is the core problem USPL would like to solve.

## 3 Method

In this section, we provide a detailed introduction to the design principles and details of USPL. Specifically, we describe the structure of the privileged and deployment policies in Sec. 3.1, explain the sampling and training processes in Sec. 3.2, and conclude the overall training process in Sec. 3.3.

### 3.1 Framework

The overall framework of USPL is shown in Fig. 2. USPL divides the observation information into three categories: common observation $o_c \in \mathcal{O}_c$, target observation $o_t \in \mathcal{O}_t$, and privileged observation $o_p \in \mathcal{O}_p$, where $\mathcal{O}_c, \mathcal{O}_t, \mathcal{O}_p$ denote the observation spaces. Privileged observation is only available during training and includes data that cannot be directly accessed during deployment, such as the coordinates of the goal points and the parameters of the environment dynamics. Common observation refers to the observation information that is available both during training and deployment, such as the joint angles, velocities, and positions of a quadruped robot. Target observation represents

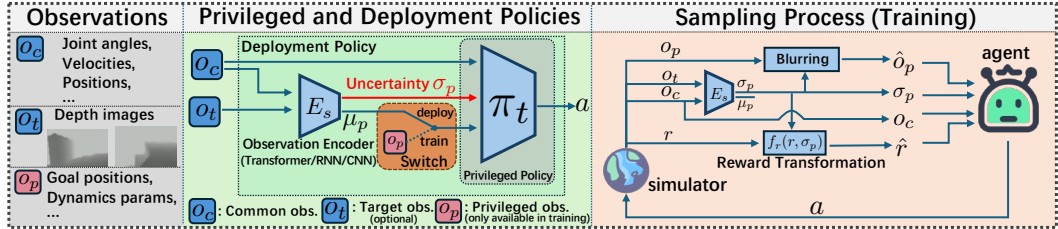

Figure 2: Overview of USPL.

high-dimensional observations that can be used to infer privileged observations during deployment, such as depth images captured by a depth camera. This type of observation is optional, not necessary in all environments, depending on the task requirements. During training, we can access all types of observations, but during deployment, only common and target observations are accessible. Therefore, our goal is to efficiently obtain a deployment policy being able to solve the task based solely on common and target observations.

USPL consists of two models: the privileged policy represented by an MLP and an observation encoder that predicts the privileged observation. The observation encoder predicts the current privileged observation based on the common and target observations. Since target observations are not globally available and may be images, the observation encoder is typically represented by a CNN-Transformer architecture to handle both image and historical memory, or a Transformer structure to process purely historical memory. The observation encoder outputs a mean $\mu_p$ and standard deviation $\sigma_p$, forming a distribution of the privileged observation, $\mathcal{N}(\cdot \mid \mu_p, \sigma_p^2)$. This distribution represents the confidence interval or feasible range for the current privileged observation based on past target observations. The standard deviation $\sigma_p$ provides an estimate of the uncertainty in the predicted privileged observation: the larger the $\sigma_p$, the less reliable the output $\mu_p$, also indicating that the DP has yet to gather sufficient privileged information.

The privileged policy is represented by a simpler MLP network, which takes as input the common observation, privileged observation, and the uncertainty output by the observation encoder, and then outputs an action. In the deployment scenario, we treat the combination of the observation encoder and the privileged policy as the deployment policy, where the privileged observation is replaced by the $\mu_p$ predicted by the observation encoder and fed into the privileged policy. The key difference from previous works is that we input the uncertainty into the privileged policy, allowing it to directly and dynamically sense the uncertainty of the observation encoder in the current state. This enables the privileged policy to intelligently identify which dimensions are inaccurate based on this uncertainty and take corresponding information-gathering actions to minimize the uncertainty, ensuring that the privileged observation can be accurately restored during deployment.

## 3.2 Training Process

The observation encoder is optimized and updated through a simple maximum likelihood estimation (MLE). Let the parameters of the observation encoder $E_s(o_c, o_t; \phi)$ be denoted as $\phi$, and for convenience, let $E_s^\mu(o_c, o_t; \phi)$ and $E_s^\sigma(o_c, o_t; \phi)$ represent the mean and standard deviation vectors output by the observation encoder, respectively. Then, given a dataset constructed by $\mathcal{D} = \{(o_c, o_t, o_p)\}$, the loss function for $\phi$ is given by the following MLE loss:

$$\mathcal{L}_{MLE}(\phi, \mathcal{D}) = -\mathbb{E}_{o_c, o_t, o_p \sim \mathcal{D}} \left[ \log \mathcal{N}(o_p | E_s^\mu(o_c, o_t; \phi), (E_s^\sigma(o_c, o_t; \phi))^2) \right]. \tag{1}$$

The training data $\mathcal{D}$ consists of two parts: one half comes from on-policy data collected by $\pi_t$, while the other half comes from past training data stored in a replay buffer. The inclusion of off-policy data is intended to enhance the diversity of the observation encoder's training dataset, helping to prevent the observation encoder from overfitting to the behavior of the current privileged policy.

The privileged policy is trained using PPO. To ensure that the deployment policy can also complete the task, we want the privileged policy to learn such a behavior pattern: when uncertainty is high, it should avoid executing behaviors that rely on the privileged observation to reduce uncertainty, thus allowing the observation encoder's output to converge to the actual privileged observation. When uncertainty is low, the privileged policy should efficiently complete the task based on the privileged observation.

This way, the deployment policy can consistently align with the behavior of the privileged policy. To achieve this, as shown in the right part of Fig. 2, we introduce two modifications during PPO sampling: reward modification and privileged blurring. The reward modification shares similarity with previous works [20], where the reward of the privileged policy is adjusted based on the difference between the privileged policy and the deployment policy. Unlike earlier methods [20] which applied an additive penalty to the original reward, we propose the following reward transformation function:

$$\hat{r} = f_r(r, \sigma_p) = [\text{clip}(C_{\min} - \log(\bar{\sigma}_p), 0, C_{\max}) + \delta] \cdot r. \tag{2}$$

Here, $C_{\min}$ and $C_{\max}$ are two hyper-parameters, such as $C_{\min} = \log(0.2)$ and $C_{\max} = C_{\min} - \log(0.01)$. $\bar{\sigma}_p$ represents the average over each dimension of $\sigma_p$, and $\delta$ is a small value that prevents the reward from being zero. This function scales the original reward, giving a lower scale coefficient for high uncertainty rewards and a higher scale coefficient for low uncertainty rewards. When uncertainty is high, we want to encourage the privileged policy to explore the environment, which may conflict with the task reward's objective, potentially leading to lower or even negative rewards. In this case, we handle the reward with a lower scale to reduce the penalty for exploration. In contrast, when uncertainty is low, we encourage the privileged policy to focus on completing the task, thus giving it a higher scale to focus on task completion.

In addition to reward modification, we inject noise into the privileged observation based on uncertainty. For continuous-valued privileged observations, we process them as follows:

$$\hat{o}_p = f_o^c(o_p, \sigma_p) = o_p + K\sigma_p\epsilon_c, \tag{3}$$

where $K$ is a hyperparameter that controls the magnitude of the noise perturbation, and $\epsilon_c$ is a random noise. The choice of distribution for $\epsilon_c$ primarily depends on the properties of $o_p$: for bounded $o_p$, we adopt a uniform distribution $\mathcal{U}(-1, 1)$ as the distribution for $\epsilon_c$. After the distribution is scaled by $K$, we clip its upper and lower bounds to ensure that the range of $\hat{o}_p$ remains within the valid bounds of $o_p$. For unbounded $o_p$, a Gaussian distribution $\mathcal{N}(\epsilon_c|0, I)$ can also be used as the noise distribution. For discrete-valued privileged observations, the perturbation is handled as follows:

$$\hat{o}_p = f_o^d(o_p, \sigma_p) = \mathbb{I}\{\epsilon_d < \sigma_p\}\tilde{o}_p + \mathbb{I}\{\epsilon_d \geq \sigma_p\}o_p, \tag{4}$$

where $\epsilon_d \sim \mathcal{U}(0, 1)$, $\mathbb{I}\{\cdot\}$ is an indicator function, and $\tilde{o}_p \sim \mathcal{U}(\mathcal{O}_p)$ is selecting uniformly at random from $\mathcal{O}_p$. Blurring the privileged observations has several advantages. Firstly, it allows the privileged policy to behave without relying on privileged observations. Secondly, during high uncertainty, the noise added to the privileged observation encourages the privileged policy to explore the environment more. Lastly, this approach helps make the privileged policy more robust to noise in the privileged observation, enabling it to tolerate errors from the observation encoder during deployment.

### 3.3 Practical Algorithm

The entire algorithmic process is summarized in Alg. 1 in App. A.1. In each iteration, we first collect the common, target, and privileged observations, denoted as $o_c$, $o_t$, and $o_p$, respectively, from the environment. Next, the observation encoder is used to compute the current uncertainty $\sigma_p$. The privileged observation $o_p$ is then perturbed using Eq. 3 or Eq. 4 to obtain $\hat{o}_p$. The privileged policy is invoked to generate the action output $a$ based on $(\hat{o}_p, o_c, \sigma_p)$. After executing the action, the next set of observations and the reward are received from the environment. The reward is scaled using Eq. 2 to obtain $\hat{r}$. Finally, the state transition and $\hat{r}$ are inserted into both the on-policy and off-policy buffers. After sample collection is complete, we train the policy using PPO [37] with the $(\hat{o}_p, o_c, \hat{r}, a)$ samples from the on-policy buffer. Additionally, the observation encoder is optimized using the $o_p, o_c, o_t$ samples from both the on-policy and off-policy buffers in conjunction with Eq. 1. More implementation details can be found in App. C.3.

### 3.4 Theoretical Analysis

In this subsection, we analyze the distribution divergence between PP and DP of USPL. Let the USPL policy be denoted as $\pi(\cdot \mid o_c, o_p, \sigma_p)$, where $o_c$, $o_p$, and $\sigma_p$ represent the common observation, privileged observation, and uncertainty estimation, respectively. At any timestep $h$, PP takes the true privileged observation $o_p^h$ as its input, while the DP uses a predicted observation $\hat{o}_p^h$ as the input.

**Theorem 1.** *Assume for each timestep $h \in \{1, \ldots, H\}$, there exists $\epsilon^h > 0$ such that the prediction error is bounded:*

$$\left\|o_p^h - \hat{o}_p^h\right\|_2 \leq \epsilon^h. \tag{5}$$

*Further assume the policy $\pi$ is Lipschitz continuous with respect to $o_p$, such that for any $o_p^h, \hat{o}_p^h$:*

$$D_{TV}\big(\pi(\cdot \mid o_c, o_p^h, \sigma_p), \pi(\cdot \mid o_c, \hat{o}_p^h, \sigma_p)\big) \leq L(o_c, \sigma_p) \, \|o_p^h - \hat{o}_p^h\|_2, \tag{6}$$

*where $L(o_c, \sigma_p)$ is the Lipschitz constant. Let $\pi_h^{\mathrm{PP}} = \pi(\cdot \mid o_c, o_p^h, \epsilon^h)$ and $\pi_h^{\mathrm{DP}} = \pi(\cdot \mid o_c, \hat{o}_p^h, \epsilon^h)$. Then, the TV distance between the state visitation distributions $d_h^{\pi^{\mathrm{PP}}}(o_c)$ and $d_h^{\pi^{\mathrm{DP}}}(o_c)$ is bounded:*

$$\frac{1}{H} \sum_{h=1}^{H} D_{TV}\left(d_h^{\pi^{PP}}, d_h^{\pi^{DP}}\right) \leq \sum_{h=1}^{H} \mathbb{E}_{o_c \sim d_h^{\pi^{PP}}}\left[L(o_c, \epsilon^h)\, \epsilon^h\right]. \tag{7}$$

*Proof.* Please refer to App. B. $\qquad\square$

Theorem 1 indicates that the divergence is governed by two factors: the prediction error $\epsilon^h$ and the Lipschitz constant $L(o_c, \epsilon^h)$, which quantifies the policy's reliance on the privileged observation $o_p$. As the true error bound $\epsilon^h$ is inaccessible in practice, we utilize the uncertainty estimation $\sigma_p$ as its approximation. The bound suggests that the divergence can be minimized by reducing either $\epsilon^h$ or $L(o_c, \epsilon^h)$. In USPL, the reward transformation (Eq. 2) is designed to minimize $\epsilon^h$, while the privileged observation perturbation (Eqs. 3 and 4) aims to minimize $L(o_c, \epsilon^h)$.

In practice, neither term can be guaranteed to remain minimal throughout an episode. Insufficient information, particularly early in an episode, often leads to high prediction error (large $\epsilon^h$). Similarly, $L(o_c, \epsilon^h)$ cannot be persistently minimized, as the DP must utilize $o_p$ at certain timesteps to maximize rewards; otherwise, the training of DP would be very difficult. A large divergence arises when both $\epsilon^h$ and $L(o_c, \epsilon^h)$ are simultaneously high. A core contribution of USPL is to make the policy uncertainty-sensitive by providing the uncertainty estimate $\sigma_p$ (as an approximation of $\epsilon^h$) as an explicit input. This enables the policy to learn a dynamic strategy: it learns to minimize its reliance on $o_p$ (lowering $L(o_c, \epsilon^h)$) when uncertainty is high, and conversely, to leverage $o_p$ to complete the task when uncertainty is low.

## 4 Experiments

In this section, we answer the following questions through experiments:

- What is the behavioral difference between the USPL privileged policy and the deployment policy? (Fig. 4)
- How well does USPL perform in various tasks? (Tab. 1 and Fig. 5)
- Can the USPL observation encoder accurately predict uncertainty, and is the privileged policy sensitive to uncertainty? (Figs. 6 and 7)
- How do the different components of USPL impact the final performance? (Tab. 2)

### 4.1 Experiment Setup

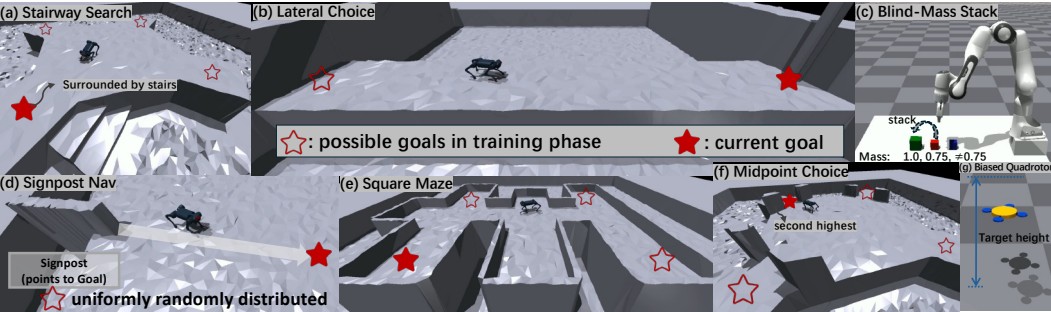

Figure 3: Illustrations of the environments used in the experiments.

We consider three common types of robots in Isaacgym [38]: a quadruped robot, a quadrotor UAV, and a robotic arm. As shown in Fig. 3, we constructed 7 environments based on these three types of

robots, with two of them supporting both image-based and non-image-based inputs, resulting in a total of 9 tasks. These tasks all require the agent to possess strong information-gathering abilities and the capacity to handle long-term memory and images. For example, in the Midpoint Choice task, the quadruped robot needs to identify the second tallest of three pillars of different heights. However, it can only perceive a small area in front of it, so it must approach each pillar, remember its height, and finally compare to determine the position of the second tallest pillar. The agent not only needs to actively explore the environment but also remember what it saw during its movement. In the Signpost Navigation task with depth image observation, the quadruped robot should first approach a signpost, use the depth camera to determine the position of the target point, and then remember this position while gradually moving toward the target. This requires processing images while maintaining long-term memory. For more detailed environment descriptions, refer to App. C.1.

We consider four baseline methods: (1) RMA [13], a privileged learning method, which differs from USPL in that it does not apply any regularization to the training of the privileged policy; (2) RESeL-PPO, which applies the state-of-the-art POMDP RL method RESeL [25] to the PPO algorithm [37], training the deployment policy purely via RL; (3) PrivRecons, a hybrid method combining RL and privileged learning, which samples based on the deployment policy but only uses RL to update the privileged policy part, while updating the observation encoder via supervised learning [39, 33]. (4) RewReg, a privileged learning variant that penalizes the PP's rewards in states where the DP exhibits large behavior divergence, as done in [20]. Refer to App. C.2 for more baseline details.

## 4.2 Trajectory Divergence between Privileged and Deployment Policies

The central motivation of USPL is to reduce the behavioral discrepancy between the Deployment Policy (DP) and the Privileged Policy (PP). To evaluate whether this objective has been achieved, we conduct experiments on four quadruped navigation tasks, measuring the divergence in trajectories between the DP and PP. As shown in Fig. 4, USPL exhibits significantly lower trajectory discrepancies compared to RMA. Notably, in environments such as Signpost Nav, Square Maze, and Lateral Choice, the discrepancy remains consistently close to zero over time, indicating that USPL effectively minimizes behavioral differences. This suggests that the DP and PP maintain highly consistent trajectories even after long-term interaction with the environment.

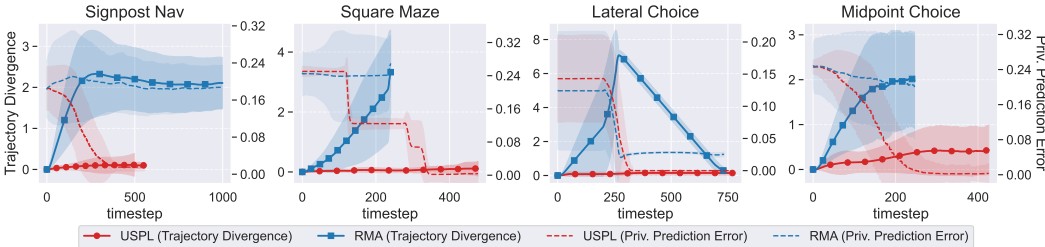

Figure 4: Trajectory divergence between PP and PD and privileged observation prediction error over time for different algorithms. The discrepancies in the lengths of the curves come from the different trajectory lengths for different algorithms.

The dashed lines in Fig. 4 represent the real-time prediction error of the observation encoder, which attempts to reconstruct the privileged observation from target observations during DP execution. Initially, when informative observations have not yet been revealed, both RMA and USPL exhibit similar levels of prediction error. However, as critical information is acquired, USPL's prediction error steadily decreases and eventually converges near zero. In contrast, RMA shows a noticeable reduction in prediction error only in the Lateral Choice environment.

From the trends in prediction error, two key conclusions emerge. First, USPL learns effective information-gathering behaviors that actively reduce the prediction error of the observation encoder during interaction. Second, even when the prediction error is high, USPL is able to maintain low behavioral discrepancies between DP and PP. This indicates that the privileged policy learns to selectively utilize privileged observations in a way that reduces the behavioral gap between the privileged and deployment policies.

## 4.3 Policy Performance Comparisons

Table 1: Success-rate comparisons over 5 seeds. "-p" denotes the privileged policies.

| Task | PrivRecons | RESeL-PPO | RewReg | RMA | USPL | RMA-p | USPL-p |
|------|-----------|-----------|--------|-----|------|-------|--------|
| Signpost Nav | 18.1±12.3 | 3.9±0.3 | 34.2±16.8 | 9.5±0.6 | **97.8±0.2** | 100.0 | 100.0 |
| Square Maze | 26.7±10.9 | 5.2±4.3 | 44.9±1.1 | 24.5±1.0 | **98.5±0.1** | 98.1 | 99.8 |
| Lateral Choice | 98.0±0.5 | 46.7±21.8 | 56.4±2.0 | 89.6±5.2 | **98.8±0.4** | 100.0 | 100.0 |
| Midpoint Choice | 80.9±3.8 | 0.1±0.1 | 50.1±1.6 | 38.8±0.4 | **97.7±0.7** | 100.0 | 98.9 |
| Biased Quadrotor | 2.2±0.0 | 0.0±0.0 | 0.6±0.2 | 0.0±0.0 | **96.7±0.1** | 100.0 | 100.0 |
| Blind-Mass Stack | 55.1±2.5 | 50.2±1.2 | 55.8±8.8 | 50.1±0.6 | **97.5±1.0** | 100.0 | 100.0 |
| Average | 46.8 | 17.7 | 40.3 | 35.4 | **97.8** | 99.7 | 99.8 |

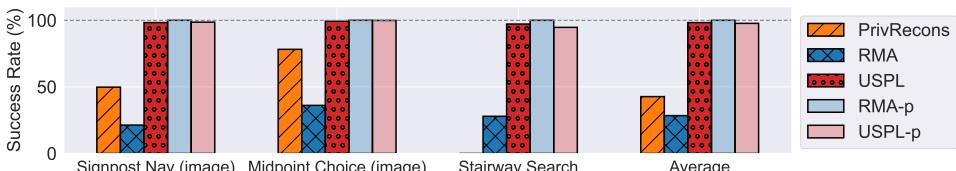

Figure 5: Success rate comparisons in image-based tasks. '-p' denotes the privileged policies.

To further quantify the performance of different algorithms during deployment, we compare the task success rates of the deployment policies across six non-visual tasks, as shown in Tab. 1. To better highlight the performance gap between privileged and deployment policies, the last two columns of the table report the task success rates of the privileged policies for both USPL and RMA. Comparing USPL with other methods, we observe that USPL consistently achieves significantly higher deployment success rates. The pure RL method (RESeL-PPO) and privileged learning methods with or without regularization on the privileged policy (RewReg and RMA) underperform the hybrid approach PrivRecons. This is primarily because the tasks heavily rely on long-term memory, rendering RL inefficient. Moreover, there exists a substantial behavioral discrepancy between the optimal privileged and deployment policies. As a result, even though RMA's privileged policy achieves over 99% average success rate, its deployment policy still performs poorly. With reward penalty regularizing the PP, RewReg outperforms RMA in 5/6 tasks but still inferior to PrivRecons, which combines RL with supervised learning. PrivRecons provides some performance improvement over other methods but lags behind USPL by up to 50%. In contrast, USPL exhibits minimal performance gap between the privileged and deployment policies, achieving over 95% success rate across all tasks. These results demonstrate the superior deployment performance of USPL in memory-intensive tasks. The learning curves of USPL and other methods can be found in App. D.1. We also compare USPL with the baselines in a non-robotic environment in App. D.2.

To further evaluate the effectiveness of USPL in high-dimensional settings, we conducted experiments on three image-based tasks. As shown in Fig. 5, USPL achieves over 95% success rates, significantly outperforming baselines such as RMA and PrivRecons. Notably, the performance gap between USPL's deployment policy and its privileged policy remains minimal, with a much smaller drop compared to RMA.

## 4.4 Uncertainty Estimation Analysis

To further understand how uncertainty estimation helps reduce the behavioral divergence between the privileged policy and the deployment policy, we conducted additional experiments. In Fig. 6, we visualize the temporal evolution of the deployment policy's uncertainty estimation in the Stairway Search task, along with the corresponding policy behaviors and the distribution of privileged observations predicted by the observation encoder at key time steps. Initially, the uncertainty estimation is around 2.0, and the predicted privileged observation distribution forms a Gaussian centered in the middle of the map with a wide spatial spread. As the robot approaches the edge of the platform and observes no stairs on that side, the uncertainty estimation drops for the first time, and the distribution $\mathcal{N}(\cdot \mid \mu_p, \sigma_p^2)$ quickly shifts toward the lower part of the map. Upon reaching the lower edge and looking to the right, the robot again finds no stairs, prompting a second drop in uncertainty—now suspecting that the stairs might be in the bottom-left corner. At this point, $\mathcal{N}(\cdot \mid \mu_p, \sigma_p^2)$ centers around the bottom-left,

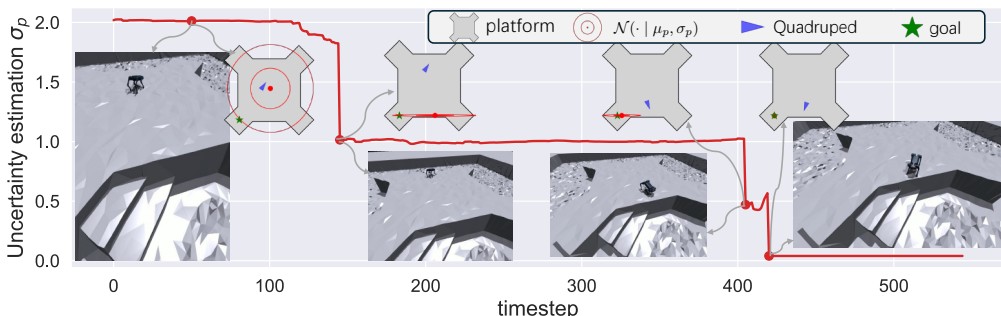

Figure 6: The uncertainty estimation of the deployment policy over timesteps, with visualizations of the corresponding behaviors and the distribution $\mathcal{N}(\cdot \mid \mu_p, \sigma_p^2)$ at key timesteps.

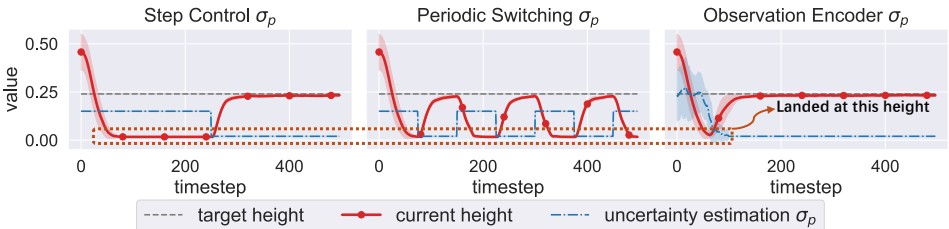

Figure 7: Altitude trajectories under different uncertainty inputs in the Biased Quadrotor task. The quadrotor aims to maintain level flight but receives biased attitude and altitude observations. During deployment, it must first land on plat ground to calibrate its sensors. The privileged observation provides the ground-truth bias.

though $\sigma_p$ does not collapse to zero since the agent has not yet directly observed the stairs. Once the robot turns to the bottom-left and detects the stairs, the uncertainty estimation immediately collapses to zero, and $\mathcal{N}(\cdot \mid \mu_p, \sigma_p^2)$ converges to the target location. This result indicates that our observation encoder provides a highly sensitive estimation of uncertainty, which accurately reflects the amount of information still to be gathered—perfectly aligning with our expectations.

To better examine how the privileged policy responds to uncertainty estimation, we further conduct experiments on the Biased Quadrotor task. We directly feed the ground-truth privileged observation into the privileged policy and vary the uncertainty estimation across three settings. The resulting altitude trajectories are shown in Fig. 7. Under the step control setting, we apply a high uncertainty estimation for the first 250 steps, then switch to a low value. Remarkably, although the policy receives the ground-truth privileged observation, it remains grounded for the first 250 steps, refusing to initiate altitude control. This is because the policy considers the calibration incomplete, and any premature action might diverge from the deployment policy. Once the uncertainty drops after step 250, the quadrotor immediately takes off and stabilizes at the desired altitude. Notably, the reward remains negative while grounded, yet the policy prioritizes reducing behavioral divergence by waiting for uncertainty to decrease. Similar behavior is observed under periodic switching and when using encoder-predicted $\sigma_p$: the policy remains grounded during high uncertainty for calibration, and takes off only when uncertainty becomes low. These results show that the USPL privileged policy is highly sensitive to uncertainty estimation and adapts its behavior accordingly.

Taken together, Figs. 6 and 7 illustrate how USPL works. The observation encoder provides rapid and accurate uncertainty predictions, while the privileged policy reacts sensitively to these signals—avoiding reliance on inaccurate privileged observations under high uncertainty and leveraging them effectively under low uncertainty to accomplish the task. Additional results similar to those in Figs. 6 and 7, covering more environments, are presented in App. D.3.

### 4.5 Ablation Studies

USPL consists of three core components: using uncertainty estimation as an input to the privileged policy (`UI`), applying blurring to the privileged observations (`blur`), and modifying the reward

Table 2: Ablation studies on uncertainty estimation input (`UI`), privileged blurring (`blur`), and reward transformation (`rew`) over 3 seeds.

| | USPL | w/o `blur` | w/o `rew` | w/o `UI` | only `UI` | RMA |
|---|---|---|---|---|---|---|
| Signpost Nav | **97.8 ± 0.2** | 96.5 ± 0.8 | 35.2 ± 1.0 | 97.0 ± 0.1 | 9.9 ± 0.4 | 9.5 ± 0.6 |
| Square Maze | **98.5 ± 0.1** | 38.9 ± 6.1 | 88.6 ± 8.3 | 94.7 ± 2.2 | 24.9 ± 0.4 | 24.5 ± 1.0 |
| Lateral Choice | **98.8 ± 0.4** | 88.1 ± 4.5 | 91.1 ± 3.5 | 92.0 ± 0.8 | 68.9 ± 2.1 | 89.6 ± 5.2 |
| Midpoint Choice | **97.7 ± 0.7** | 50.9 ± 5.8 | 94.4 ± 1.1 | 78.9 ± 0.6 | 0.0 ± 0.0 | 38.8 ± 0.4 |
| Avg. Success Rate | **98.2** | 68.6 | 77.3 | 90.7 | 25.9 | 40.6 |
| Avg. Failure Rate | **1.8** | 31.4 | 22.7 | 9.3 | 74.1 | 59.4 |

function (`rew`). To investigate the contribution of each component, we conducted ablation and sensitivity studies, with the ablation results shown in Tab. 2, and the sensitivity results in App. D.4. We found that both `blur` and `rew` significantly impact policy exploration. When both are removed (i.e., only `UI`), performance drops sharply across all four environments, as the PP lacks proper guidance and fails to minimize behavioral divergence from the DP. The effects of `blur` and `rew` vary by task: in the Signpost Nav task, `rew` is crucial—its removal leads to a major performance drop—while `blur` has little effect. In contrast, for the other three tasks, removing `blur` causes substantial degradation, whereas removing `rew` has a smaller impact. Combining both consistently yields strong performance across all tasks. The results show that removing `UI` still results in reasonable performance under the strong guidance of `rew` and `blur`, but it falls short of optimal. We found that introducing `UI` significantly reduces the policy's failure rate. `UI` gains an $81\%$ relative reduction in the failure rate (comparing USPL to w/o `UI`), substantially improving policy robustness.

Tab. 2 demonstrates how different components play distinct yet complementary roles in a multi-stage process. Note that an optimal policy must adapt its behavior online based on its uncertainty estimation, $\sigma_p$. USPL achieves this in two stages. First, the policy must learn a complete spectrum of optimal behaviors corresponding to different uncertainty levels. This is achieved via two key components: Privileged blurring simulates varying error levels in the privileged observation, thereby training the agent to master behaviors suited for different reliability levels of this input. Simultaneously, the reward transformation incentivizes the agent to learn exploratory behaviors that actively minimize uncertainty. Tab. 2 confirms that both components are essential for discovering this full behavioral spectrum. Second, the policy must dynamically select the appropriate behavior online. Providing $\sigma_p$ as a direct input acts as a real-time signal, enabling the policy to seamlessly switch between exploratory behaviors (high $\sigma_p$) and exploitative behaviors that utilize the privileged observation for task execution (low $\sigma_p$). This online adaptation fails if the behavioral spectrum was not learned in the first stage, which explains the poor performance observed when only providing the uncertainty input.

## 5 Conclusions

In this paper, we propose Uncertainty-Sensitive Privileged Learning (USPL) to address the limitation that the privileged policy (PP) in prior methods lacks real-time awareness of the progress of privileged information gathering. USPL constructs the deployment policy (DP) by combining the privileged policy (PP) with an observation encoder that predicts privileged observations and outputs an associated estimate of uncertainty. The uncertainty, which reflects how much information still needs to be gathered, is fed to the PP so it can adapt its behavior as information gathering progresses. Additionally, USPL encourages the PP to produce DP-replicable behaviors via reward transformation and privileged observation blurring. Experiments show that the DP can accurately estimate uncertainty, the PP learns to actively explore and calibrate, and the behavior divergence between PP and DP is largely reduced, achieving over 95% deployment success rate on all 9 tasks.

**Limitations.** Despite the promising results, USPL still has two known limitations: (1) It currently handles only low-dimensional privileged observations. Extending USPL to high-dimensional, redundant, or time-varying privileged observations remains an open challenge. It may be addressed by introducing a privileged encoder that maps privileged observations into a latent space [2, 13]. (2) Another potential direction is to eliminate the use of the DP's encoder during PP training, thereby avoiding the need to process high-dimensional target observation information and thus improving both training and simulation speed. One possible solution is to train a proxy encoder that directly predicts uncertainty from historical privileged observations [20].

## Acknowledgments and Disclosure of Funding

This work is supported by the National Science Foundation of China (62495093,62506159, U24A20324), the Natural Science Foundation of Jiangsu (BK20241199, BK20243039), and the AI & AI for Science Project of Nanjing University. The authors thank Dr. Tian Xu for his help with the theoretical derivations. The authors thank the anonymous reviewers for their support and helpful discussions on improving the paper.

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

# A    Algorithm Details

## A.1    Algorithm

---

**Algorithm 1** Uncertainty-Sensitive Privileged Learning (USPL)

---

1: Initialize $\pi_t$, $E_s$, on-policy buffer $\mathcal{B}_{\text{on}}$ and off-policy buffer $\mathcal{B}_{\text{off}}$;
2: **for** each training iteration **do**
3:     Reset simulator, obtain common, target, and privileged observations $o_c, o_t, o_p, \mathcal{B}_{\text{on}} \leftarrow \emptyset$;
4:     **for** each step **do**
5:         $\mu_p, \sigma_p \leftarrow E_s(o_c, o_t; \phi)$;
6:         $\hat{o}_p \leftarrow f_o(o_p, \sigma_p)$ following Eq. 3 or Eq. 4 based on the continuity of $o_p$;
7:         $a \leftarrow \pi_t(\hat{o}_p, o_c, \sigma_p)$;
8:         Apply $a$ to the simulator, obtain reward $r$ and next observations $o_c', o_t', o_p'$;
9:         $\hat{r} \leftarrow f_r(r, \sigma_p)$ following Eq. 2;
10:        Store $(o_t, \hat{o}_p, o_p, o_c, a, r, o_c', o_t', o_p')$ in both $\mathcal{B}_{\text{on}}$ and $\mathcal{B}_{\text{off}}$;
11:    **end for**
12:    Update $\pi_t$ using PPO with $\mathcal{B}_{\text{on}}$;
13:    Update $E_s$ using Eq. 1 with $\mathcal{B}_{\text{off}}$ and $\mathcal{B}_{\text{on}}$;
14: **end for**

---

# B    Proof

## B.1    Proof Preliminaries

**Lemma 1** (Lemma 4 in [40])**.** *For any tabular and episodic MDP, considering two policies $\pi$ and $\pi'$, let $d_h^\pi(\cdot)$ denote the state distribution induced by $\pi$ in time step $h$. When $h \geq 2$, we have that*

$$\|d_h^\pi(\cdot) - d_h^{\pi'}(\cdot)\|_1 \leq \sum_{\ell=1}^{h-1} \mathbb{E}_{s \sim d_\ell^{\pi'}(\cdot)}[\|\pi_\ell(\cdot \mid s) - \pi_\ell'(\cdot \mid s)\|_1]. \tag{8}$$

*Proof.* Please refer to [40]. $\qquad\square$

**Corollary 1.** *For any tabular and episodic MDP, considering two policies $\pi$ and $\pi'$, let $d_h^\pi(\cdot)$ denote the state distribution induced by $\pi$ in time step $h$. If there exists $\varepsilon \in \mathbb{R}$ such that*

$$\frac{1}{H} \sum_{\ell=1}^{H} \mathbb{E}_{s \sim d_\ell^{\pi'}} \left[ D_{\text{TV}}\big(\pi_\ell(\cdot \mid s), \pi_\ell'(\cdot \mid s)\big) \right] \leq \varepsilon, \tag{9}$$

*then*

$$\frac{1}{H} \sum_{h=1}^{H} D_{\text{TV}}\big(d_h^\pi, d_h^{\pi'}\big) \leq H \varepsilon. \tag{10}$$

*Proof.* We adopt the standard definition of the TV distance:
$$D_{\text{TV}}(p, q) := \tfrac{1}{2}\|p - q\|_1.$$
Thus, Eq. 8 can be written as follows:

$$D_{\text{TV}}\big(d_h^\pi, d_h^{\pi'}\big) \leq \sum_{\ell=1}^{h-1} \mathbb{E}_{s \sim d_\ell^{\pi'}} \left[ D_{\text{TV}}\big(\pi_\ell(\cdot \mid s), \pi_\ell'(\cdot \mid s)\big) \right].$$

Summing over $h = 1, \ldots, H$ and exchanging the order of summations,

$$\sum_{h=1}^{H} D_{\text{TV}}\big(d_h^\pi, d_h^{\pi'}\big) \leq \sum_{\ell=1}^{H-1} (H - \ell) \, \mathbb{E}_{s \sim d_\ell^{\pi'}} \left[ D_{\text{TV}}\big(\pi_\ell(\cdot \mid s), \pi_\ell'(\cdot \mid s)\big) \right]$$

$$\leq H \sum_{\ell=1}^{H} \mathbb{E}_{s \sim d_\ell^{\pi'}} \left[ D_{\text{TV}}\big(\pi_\ell(\cdot \mid s), \pi_\ell'(\cdot \mid s)\big) \right].$$

Divide both sides by $H$ and apply Eq. 9 to obtain the corollary. $\qquad\square$

## B.2 Proof of Theorem 1

*Proof.* We first bound the single-step policy divergence at timestep $h$ for a given common observation $o_c$. By the definitions of $\pi_h^{\text{PP}}$ and $\pi_h^{\text{DP}}$, the Lipschitz continuity assumption in Eq. 6, and the prediction error bound in Eq. 5, we have:

$$
\begin{aligned}
D_{\text{TV}}\big(\pi_h^{\text{PP}}(\cdot \mid o_c), \pi_h^{\text{DP}}(\cdot \mid o_c)\big) &= D_{\text{TV}}\Big(\pi(\cdot \mid o_c, o_p^h, \epsilon^h),\, \pi(\cdot \mid o_c, \hat{o}_p^h, \epsilon^h)\Big) \\
&\leq L(o_c, \epsilon^h)\, \|o_p^h - \hat{o}_p^h\|_2 \\
&\leq L(o_c, \epsilon^h)\, \epsilon^h.
\end{aligned}
\tag{11}
$$

Taking the expectation of this inequality with respect to $o_c \sim d_h^{\pi^{\text{PP}}}$ gives:

$$
\mathbb{E}_{o_c \sim d_h^{\pi^{\text{PP}}}}\left[ D_{\text{TV}}\big(\pi_h^{\text{PP}}(\cdot \mid o_c), \pi_h^{\text{DP}}(\cdot \mid o_c)\big) \right] \leq \mathbb{E}_{o_c \sim d_h^{\pi^{\text{PP}}}}\left[ L(o_c, \epsilon^h)\, \epsilon^h \right].
\tag{12}
$$

From Corollary 1, we have the following bound on the average state visitation divergence, which relates it to the sum of expected single-step policy divergences:

$$
\frac{1}{H} \sum_{h=1}^{H} D_{\text{TV}}\left( d_h^{\pi^{\text{PP}}}, d_h^{\pi^{\text{DP}}} \right) \leq \sum_{h=1}^{H} \mathbb{E}_{o_c \sim d_h^{\pi^{\text{PP}}}}\left[ D_{\text{TV}}\big(\pi_h^{\text{PP}}(\cdot \mid o_c), \pi_h^{\text{DP}}(\cdot \mid o_c)\big) \right].
\tag{13}
$$

Finally, substituting the bound from Eq. 12 into the right-hand side of Eq. 13 yields the claimed inequality:

$$
\frac{1}{H} \sum_{h=1}^{H} D_{\text{TV}}\big( d_h^{\pi^{\text{PP}}}, d_h^{\pi^{\text{DP}}} \big) \leq \sum_{h=1}^{H} \mathbb{E}_{o_c \sim d_h^{\pi^{\text{PP}}}}\left[ L(o_c, \epsilon^h)\, \epsilon^h \right].
\tag{14}
$$

$\square$

## C Experiment Settings

### C.1 Tasks Descriptions

#### C.1.1 Stairway Search

**Observation space:** depth images from an onboard depth camera, current robot position, and orientation.

**Action space:** desired change in heading (yaw) and desired body pitch.

**Privileged observation:** coordinates of the target platform.

**Reward:** a shaped reward for approaching the target platform and a terminal reward for reaching it.

**Task description.** The robot starts on a large platform and must step onto a smaller platform equipped with stairways on both sides so that it can later descend to the ground. To focus the policy on high-level decision making, the agent does not command the torques of all 12 joints directly. Instead, we train a low-level controller—based on the open-source implementation of [15]—that receives three commands: target speed magnitude, desired body pitch, and desired yaw. The controller's inputs include motor positions and velocities, the base linear and angular velocities, pitch and roll angles, the robot's inertial parameters, and the three command signals. Because precise speed control is not critical here, the commanded speed is fixed at $0.5\,\text{m/s}$.

**Optimal behavior.** The robot should peer over the platform's edge to identify the smaller platform that has stairs on both sides, then walk onto that platform.

#### C.1.2 Lateral Choice

**Observation space:** current robot position and orientation.

**Action space:** desired change in heading (yaw).

**Privileged observation:** coordinates of the goal point.

**Reward:** a shaped reward for approaching the goal and a terminal reward for reaching it.

**Task description.** The goal may appear either on the left or on the right, but the robot can perceive only its own pose—it has no information about the terrain. Under this *blind* setting the robot must navigate to the goal; the episode terminates once the goal is reached.

**Optimal behavior.** As illustrated in Fig. 1, the robot first walks toward one side. If the episode does not terminate, that side is not the goal, so the robot turns around and heads to the opposite side.

### C.1.3 Blind-Mass Stack

**Observation space:** positions of the three cubes; end-effector position; index of the green cube; boolean flag indicating whether a cube is grasped; index of the grasped cube (if any); measured mass of the grasped cube.

**Action space (three discrete dimensions):**

- end-effector $x, y$ target—discretized into three values, each corresponding to the $x, y$ position of one cube (the end-effector moves toward the selected cube);

- end-effector $z$ target—discretized into three height levels;

- gripper command—open or close.

**Privileged observation:** index of the red cube and the bias applied to the weight sensor.

**Reward.** the reward for the end-effector approaching the red block, the reward for the end-effector lifting the red block, the reward for approaching the green block while the red block is still lifted, and the reward for task completion.

**Task description.** Three cubes—red, green, and blue—are placed from left to right and numbered $1, 2, 3$. The green cube (index known) weighs $1\,\mathrm{kg}$; the red cube weighs $0.75\,\mathrm{kg}$, but its index is unknown; the blue cube's weight and index are both unknown. The arm's weight sensor is biased by a fixed multiplicative factor. The objective is to identify the red cube via weight sensing and stack it on top of the green cube, implicitly learning to self-calibrate the sensor.

**Optimal behavior.** Because the green cube's index and true mass are known, the arm first grasps the green cube to estimate the bias factor and calibrate the sensor. It then picks either remaining cube, checks whether its (calibrated) mass is $0.75\,\mathrm{kg}$; if so, that cube is red and should be stacked on the green cube, otherwise the other cube is stacked.

### C.1.4 Signpost Nav

**Observation space:** current robot position and orientation, plus either (i) terrain heights in a small frontal region obtained via `scandot` or (ii) depth images from a depth camera.

**Action space:** desired change in heading (yaw).

**Privileged observation:** coordinates of the goal point.

**Reward:** a shaped reward for moving toward the goal and a terminal reward for reaching it.

**Task description.** The robot must reach a hidden goal point whose location is not directly observable. Instead, a signpost encodes the goal: its orientation indicates direction, and its length encodes the distance from the signpost to the goal.

**Optimal behavior.** The robot first walks to the signpost, uses `scandot` or depth perception to infer the signpost's geometry, predicts the goal location accordingly, and finally travels to the goal.

### C.1.5 Square Maze

**Observation space:** current robot position and orientation.

**Action space:** desired change in heading (yaw).

**Privileged observation:** coordinates of the goal point.

**Reward:** a shaped reward for approaching the goal and a terminal reward for reaching it.

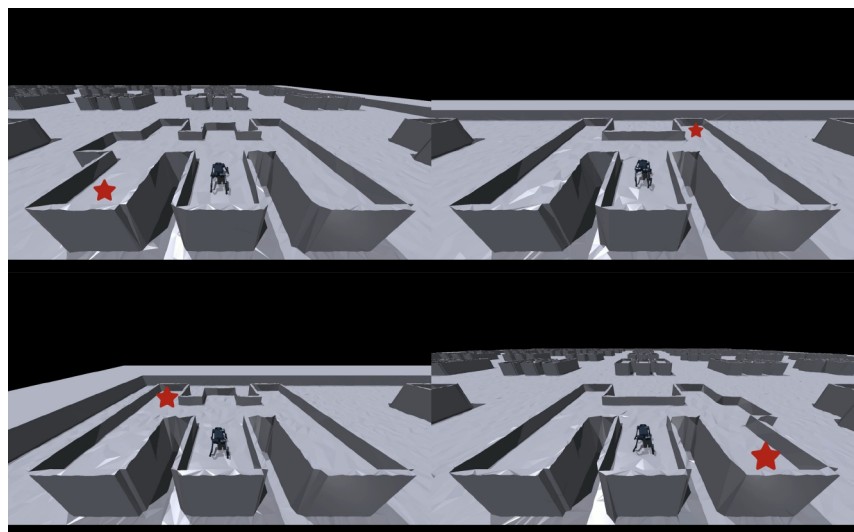

Figure 8: Visualization of the four Square Maze layouts and their corresponding goal locations.

**Task description.** The goal can be located at any of the four corners. The maze layout varies slightly with the goal location, as shown in Fig. 8. At the first junction, if the junction is a "T" shape the goal is to the right; otherwise it is to the left. A similar relationship holds at the second junction.

**Optimal behavior.** Because the robot is blind to the maze geometry, it probes each junction by walking straight ahead. If it collides with a wall, it turns right; otherwise it turns left, thereby inferring the junction type and, recursively, the goal corner.

### C.1.6  Midpoint Choice

**Observation space:** current robot position and orientation, plus either (i) terrain heights in a small frontal region (scandot) or (ii) depth images from a depth camera in front of the robot head.

**Action space:** desired change in heading (yaw).

**Privileged observation:** coordinates of the goal point.

**Reward:** a shaped reward for approaching the goal and a terminal reward for reaching it.

**Task description.** The goal resides on one of four corner platforms. Each platform is preceded by two columns, but only the goal platform has the *second-tallest* pair. With image observations, the robot can measure column heights from the depth camera; with scandot, it obtains local height data directly.

**Optimal behavior.** The robot circles the platforms, records the heights of the three column pairs it encounters, identifies the second tallest pair, and moves to the platform behind it.

### C.1.7  Biased Quadrotor

**Observation space:** perceived altitude, roll, pitch, angular velocity, vertical velocity, and a flag indicating whether the quadrotor has landed.

**Action space:** target pitch, target roll, and target altitude.

**Privileged observation:** biases in the altitude, roll, and pitch sensors.

**Reward:** a shaped reward for maintaining the target altitude and level attitude, a terminal reward for achieving them, and a penalty for flying too low (touching the ground).

**Task description.** The quadrotor must perform level hover at a target altitude, but its state estimates are corrupted by unknown sensor biases caused by an in-flight malfunction.

**Optimal behavior.** The quadrotor first lands, because ground contact provides a reference of zero altitude, roll, and pitch. Using these reference readings it calibrates its sensors, then takes off again and performs steady hover with the calibrated measurements.

Table 3: Summary of environments, observation modalities, and desired behaviors.

| Env. name | Observations | Task description | Optimal behavior |
|---|---|---|---|
| Stairway Search | common: robot pose
privileged: target coords
target: depth image | The robot must leave a large platform and step onto a smaller one with stairs on both sides. | Navigate to a platform edge, use depth perception to identify the staircase, and ascend onto the target platform. |
| Lateral Choice | common: robot pose
privileged: goal coords
target: N/A | A goal spawns either left or right, but only the robot's own pose is observable. | Walk toward one side; if the goal is not reached, turn around and head to the opposite side. |
| Blind-Mass Stack | common: cube & EE states, grasp info
privileged: red-cube index, sensor bias
target: N/A | Identify the red cube via weight sensing and stack it on the green cube while self-calibrating the arm's scale. | Calibrate on the green cube, test the others, then stack the red cube on the green one. |
| Signpost Nav | common: robot pose
privileged: goal coords
target: depth / `scandot` | A signpost encodes the hidden goal: orientation gives direction, length gives distance. | Approach the signpost, infer the goal location from its geometry, then travel to the goal. |
| Square Maze | common: robot pose
privileged: goal coords
target: N/A | The goal lies at one of four corners; junction geometry hints the correct path. | Walk straight to probe each junction; collide/clear cues determine turns that guide the robot to the goal corner. |
| Midpoint Choice | common: robot pose
privileged: goal coords
target: depth / `scandot` | Four corner platforms, three of which have two columns; the goal platform has the second-tallest pair. | Circle the platforms, measure column heights, then move to the platform behind the second-tallest pair. |
| Biased Quadrotor | common: altitude, attitude, velocities, landed flag
privileged: sensor biases
target: N/A | Hover level at a target altitude despite unknown biases in altitude and attitude sensors. | Land to obtain zero references, estimate biases, then take off and hover stably. |

We summarize the description of the environments in Tab. 3.

## C.2 Baselines

We mainly evaluate four categories of baselines:

1. **PrivRecons.** PrivRecons adopts exactly the same network architecture as USPL: the observation encoder is trained with an MLE loss, while the privileged policy is optimized with PPO. Unlike USPL, however, PrivRecons collects trajectories using the deployment

policy so that the privileged policy experiences no train–test gap. During learning, the method does not exploit privileged information as auxiliary input and is not targeted to actively gather it. It performs well only when privileged information is easy to obtain—i.e., when random exploration quickly uncovers observations correlated with that information. For tasks in which privileged information is hard to gather (or costly in reward terms) such as *Stairway Search* and *Biased Quadrotor*, the efficiency of PrivRecons drops sharply.

2. **RMA.** [13] RMA removes all innovations introduced by USPL: it neither feeds uncertainty into the privileged policy nor applies reward transformation or privileged blurring. Because the privileged policy is trained without considering the deployment setting, its performance is generally poor.

3. **RewReg.** [20] On top of RMA, RewReg adds *reward guidance* that encourages the privileged policy to minimize its behavioral divergence from the deployment policy while still completing the task. Such methods implicitly assume that the privileged policy can generate optimal behavior that is replicable during deployment. However, as discussed earlier, achieving these behaviors is challenging for the privileged policy because its input lacks information-gathering progress. RewReg outperforms RMA in most environments, but its overall performance remains far below that of USPL.

4. **RESeL_PPO.** We apply RESeL's stabilization techniques [25] to PPO training. Concretely, the deployment policy is a Transformer trained directly with PPO, and we use a reduced learning rate to stabilize Transformer optimization. Nevertheless, the tasks considered here involve long-term credit-assignment challenges. Ni et al. suggest that Transformer-based RL struggles to solve such problems efficiently [32]. As a result, RESeL_PPO still shows training inefficiency on our benchmark.

## C.3 Implementation Details of USPL

### C.3.1 More Implementation Tricks

**Learning Schedule.** In our implementation, we employ a cosine schedule to anneal both the standard deviation of the PPO policy and its learning rate. In addition, a separate cosine schedule is applied to gradually increase the degree of privileged blurring and reward transformation. Training begins with a 100-epoch warm-up phase, during which neither privileged blurring nor reward transformation is applied. This results in a policy that converges to an uncertainty-augmented RMA. The goal of this phase is to enable the privileged policy (PP) to acquire basic task-solving skills. After the warm-up, the influence of privileged blurring and reward transformation is gradually introduced following a cosine schedule. Let the progress coefficient output by this schedule be $\ell \in [0, 1]$, which increases from 0 to 1 throughout training. The reward transformation is defined as:

$$d = \left[ \mathrm{clip}(C_{\min} - \log(\bar{\sigma}_p), 0, C_{\max}) + 0.05 \right],$$
$$\hat{r} = f_r(r, \sigma_p) = \left[ (1 - d) * (1 - \ell) + d \right] \cdot r.$$

For continuous-valued privileged observations, privileged blurring is formulated as:

$$\hat{o}_p = f_o^c(o_p, \sigma_p) = o_p + \ell K \sigma_p \epsilon_c, \quad \epsilon_c \sim \mathcal{U}(-1, 1).$$

For discrete-valued privileged observations, the formulation becomes:

$$\hat{o}_p = f_o^d(o_p, \sigma_p) = \mathbb{I}\{\epsilon_d < \ell \sigma_p\}\tilde{o}_p + \mathbb{I}\{\epsilon_d \geq \ell \sigma_p\}o_p.$$

This gradual process encourages the PP to progressively reduce its reliance on privileged observations.

**Off-Policy Replay Buffer.** We use an off-policy replay buffer to train the observation encoder. To further enhance the diversity of the data stored in the buffer, we additionally collect a batch of trajectories with mismatched privileged observations during each sampling iteration. Specifically, we randomly swap privileged observations across different environments, forcing the PP to act based on incorrect privileged inputs. This strategy prevents the observation encoder from trivially inferring privileged information solely from the behavior of the PP. For example, in the Lateral Choice task, after the warm-up phase, the PP quickly converges to directly reaching the goal without exploration. As a result, the observation encoder could deduce the goal location simply by observing whether the PP moves left or right. By incorporating mismatched data, we eliminate this shortcut and promote more robust encoder training.

**Uncertainty Smoothing.** During sampling, we apply a temporal smoothing operation to the uncertainty values before feeding them into the PP. This is necessary because the uncertainty signal can be highly volatile in the early stages of training. If the uncertainty drops abruptly to a small value, the PP can immediately get the actual privileged observation and exploit it to solve the task without performing meaningful exploration—this behavior is referred to as an "uncertainty hack." By smoothing the uncertainty over time, we mitigate such exploitation and ensure that the PP continues to explore the environment effectively until sufficient information is truly gathered.

### C.3.2 Hyperparamters

Tabs. 4 and 5 list the hyperparameters used for the quadruped robot and for the other two environments, respectively. Because the trajectory lengths, observation spaces, and task complexities differ between the quadruped robot and the two additional tasks, we employ two distinct sets of hyperparameters.

Table 4: Key hyperparameters for the Quadruped Robot

| Attribute | Value |
|---|---|
| Learning rate for $E_s$ | $5 \times 10^{-5}$ |
| Weight decay for $E_s$ | 0.0 |
| Minimal reward multiplication factor ($\delta$ in Eq. 2) | 0.05 |
| Distribution of the privileged observation noise ($\epsilon_c$ in Eq. 3) | uniform distribution |
| Initial sampling standard deviation | 0.35 |
| Initial learning rate for the MLP policy of PP | $1 \times 10^{-4}$ |
| Initial learning rate for the MLP value function | $5 \times 10^{-4}$ |
| Initial learning rate for the context encoder of PP | $2 \times 10^{-5}$ |
| Initial learning rate for the context encoder of the value function | $5 \times 10^{-4}$ |
| Discount factor $\gamma$ | 0.998 |
| Number of parallel agents | 900 |
| Batch size for observation–encoder training | 8000 |
| Training gradient steps of the observation encoder per epoch | 450 |
| Training gradient steps of the privileged policy per epoch | 150 |
| Cosine schedule period of the privileged blurring and reward transformation | 300 |

Table 5: Key hyperparameters for the *Biased Quadrotor* and *Blind-Mass Stack* Tasks

| Attribute | Value |
|---|---|
| Learning rate for $E_s$ | $2 \times 10^{-5}$ |
| Weight decay for $E_s$ | $2 \times 10^{-2}$ |
| Minimal reward multiplication factor ($\delta$ in Eq. 2) | 0.05 |
| Distribution of the privileged observation noise ($\epsilon_c$ in Eq. 3) | uniform distribution |
| Initial sampling standard deviation | 0.65 |
| Initial learning rate for the MLP policy of PP | $1 \times 10^{-4}$ |
| Initial learning rate for the MLP value function | $5 \times 10^{-4}$ |
| Initial learning rate for the context encoder of PP | $2 \times 10^{-5}$ |
| Initial learning rate for the context encoder of the value function | $5 \times 10^{-4}$ |
| Discount factor $\gamma$ | 0.995 |
| Number of parallel agents | 4000 |
| Batch size for observation–encoder training | 16000 |
| Training gradient steps of the observation encoder per epoch | 170 |
| Training gradient steps of the privileged policy per epoch | 150 |
| Cosine schedule period of the privileged blurring and reward transformation | 150 |

### C.3.3 Network Architectures

In this part, we describe the architectures of the two key modules involved in USPL.

**Privileged Policy**    The architecture of the Privileged Policy is consistent with the policy design in RESeL [25], as our training framework is built upon the RESeL codebase. A context encoder, implemented as a two-layer MLP with hidden dimensions [256, 256], first encodes the environment information based on the last observation, last action, and current observation. Here, the observation includes both the common observation and the privileged observation. The context encoder outputs a 128-dimensional representation, which is then concatenated with the current observation and fed into another MLP-based policy network with hidden layers of size [256, 256]. This policy network finally outputs the action. ELU is used as the activation function for all intermediate layers.

**Observation Encoder (non-image)**    For non-image-based tasks, the observation encoder first processes the common observation through a three-layer MLP with hidden sizes [512, 256, 128]. The output is then passed to a temporal sub-module. This sub-module first applies a three-layer MLP with hidden sizes [1024, 512, 256] to extract features, followed by a Transformer block. The Transformer block is configured with 16 attention heads, each with 64 dimensions. The output of the Transformer is then processed by a 256-unit MLP to produce a 128-dimensional embedding. This embedding is further mapped to the mean and log standard deviation of the current privileged observation distribution by two MLPs with hidden sizes [512, 256]. ELU is used as the activation function in all intermediate layers, while the final embedding is activated using the Tanh function.

**Observation Encoder (image)**    In image-based tasks, the image input is first processed by a CNN encoder. The encoded image features are then concatenated with the common observation features, which are separately processed by an MLP with hidden layers [512, 256]. The combined representation is subsequently fed into the same temporal processing module as in the non-image encoder.

The CNN encoder is adapted from the implementation in [15] and consists of a convolutional layer with kernel size 5 and 32 output channels, followed by a max pooling layer with kernel size 2 and stride 2. This is followed by another convolutional layer with kernel size 3 and 64 output channels. The resulting feature map is flattened and passed through two fully connected layers, each with 128 units. ELU activation is applied to all intermediate layers, and the final embedding is activated using the Tanh function.

### C.4 Computational Infrastructure and Computational Overhead

Our experiments were conducted on GPUs with 82.6 TFLOPS of compute and 24 GB of memory. For non-image-based tasks, training was performed using a single GPU, and the training times are summarized in Tab. 6.

Table 6: Training time for non-image tasks

| Task | Time per iteration (s) | Total training time (h) |
|------|------------------------|-------------------------|
| Biased Quadrotor | 37.8 | 10.5 |
| Blind-Mass Stack | 33.0 | 18.3 |
| Lateral Choice | 47.8 | 6.6 |
| Midpoint Choice | 57.2 | 31.8 |
| Signpost Nav | 53.9 | 11.2 |
| Square Maze | 50.1 | 13.9 |

For image-based tasks, we utilized five GPUs in parallel to accelerate training. The training durations for these tasks are shown in Tab. 7.

We find that the training speed for non-image tasks is generally acceptable. However, training time for image-based tasks is significantly longer. This inefficiency is primarily caused by three factors: (1) synchronization overhead across multiple GPUs, (2) memory and GPU memory conversions when sampling batches from the replay buffer, and (3) image rendering in simulation, where environments

Table 7: Training time for image-based tasks

| Task | Time per iteration (s) | Total training time (h) |
|------|------------------------|-------------------------|
| Signpost Nav (image) | 166.9 | 34.8 |
| Midpoint Choice (image) | 177.0 | 49.0 |
| Stairway Search | 165.0 | 59.9 |

are processed sequentially, leading to considerable delays. Improving the training efficiency of image-based tasks represents a valuable and meaningful direction for future work.

Table 8: Training time of different methods on the Signpost Nav task.

| Method | Average Training Time per Epoch (s) |
|--------|-------------------------------------|
| USPL | 53.9 |
| RMA | 53.1 |
| PrivRecons | 53.2 |
| RewReg | 53.4 |
| RESeL-PPO | 46.8 |

We also compared the training time of different methods. We compared the average training time per epoch on the Signpost Nav task. The results are listed in Tab. 8. We find RESeL-PPO, which trains the entire DP network end-to-end with a PPO loss, is approximately 13% faster than the other methods that require additional loss computations. The training times for USPL and the other three baselines show no significant differences, indicating that USPL does not introduce significant computational overhead.

# D More Experiment Results

## D.1 Learning Curves

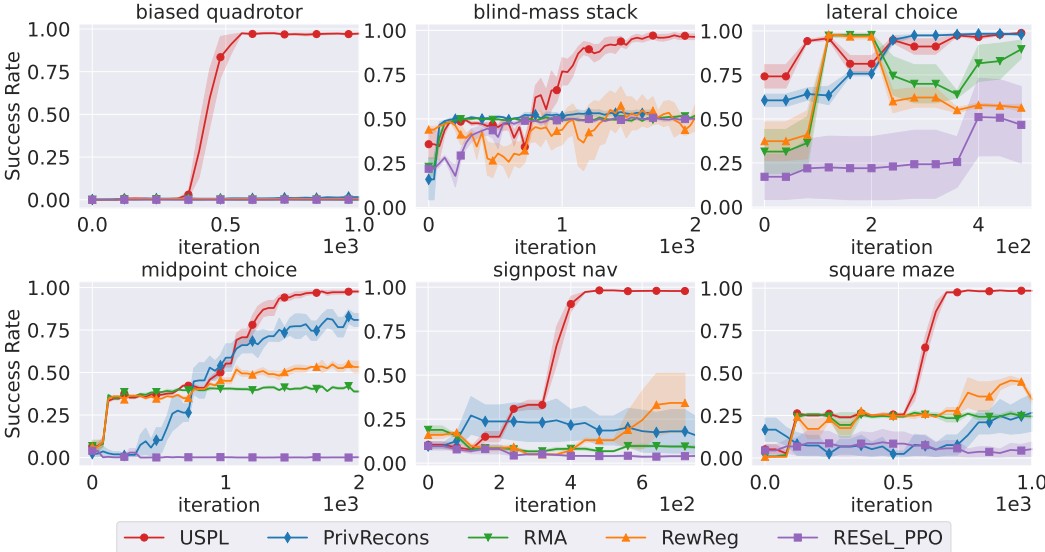

Figure 9: Success-rate curves of different algorithms over the course of training.

As a supplement to Tab. 1, Fig. 9 presents the training curves of the different algorithms. The USPL curve is notably smooth with a narrow confidence band, indicating both robustness to random initialization and high training stability. By contrast, the pure RL baseline (RESeL_PPO) exhibits

pronounced oscillations and a much wider confidence region, suggesting that pure RL suffers from instability on these tasks and yields highly variable performance across different random seeds.

## D.2 Experiments on Non-Robotic Environments

In addition to the robotic control tasks from our previous experiments, we evaluated USPL on a set of non-robotic environments. We consider the challenging credit assignment task from [32] known as Active T-Maze, setting the credit assignment length to 500. Active T-Maze is a grid-world task where the agent's navigable area is shaped like a "T" rotated 90 degrees clockwise (as in Fig. 1 of [32]). The goal is located at either the top-right or bottom-right corner. The optimal policy requires the agent to first move one step left to an "oracle" position, which reveals the goal's location. The agent must then move right to a junction and decide whether to proceed up or down based on this privileged information. [32] found that traditional Transformer-based and RNN-based RL methods struggle to solve such credit assignment tasks efficiently. We approach the Active T-Maze task from a privileged learning perspective. For this experiment, USPL was trained using 5 different random seeds, while all other baselines were trained with 3 random seeds due to limited computational resources.

Table 9: Success-rate comparisons in the Active T-Maze task. "-p" denotes the privileged policies.

|  | PrivRecons | RESeL-PPO | RewReg | RMA | USPL | RMA-p | USPL-p |
|---|---|---|---|---|---|---|---|
| Success Rate | $50.7\pm0.1$ | $43.4\pm2.1$ | $49.5\pm0.6$ | $51.4\pm0.2$ | $\mathbf{100.0\pm0.0}$ | 100.0 | 100.0 |

As shown in Tab. 9, USPL achieves a perfect success rate on this highly challenging task, whereas the other baselines only reach approximately a 50% success rate. We hypothesize that USPL's explicit uncertainty input and reward transformation mechanisms significantly alleviate the difficulty of this long-horizon credit assignment. These results demonstrate USPL's effectiveness in solving challenging credit assignment problems.

## D.3 More Uncertainty Analysis Results

### D.3.1 More Uncertainty Visualizations

As an extension to Fig. 6, we visualize in Figs. 10 to 14 the evolution of agent behavior and the predicted privileged-observation distribution over time in the *Lateral Choice*, *Square Maze*, *Midpoint Choice (image)*, *Blind-Mass Stack*, and *Biased Quadrotor* tasks, respectively. These figures reinforce the finding of Fig. 6: the uncertainty produced by the observation encoder provides a sensitive indicator of information-gathering progress. Once the key information has been collected, this uncertainty contracts rapidly, allowing the policy to adjust its behavior promptly in response to the current uncertainty level.

### D.3.2 Manually Setting Uncertainties

Complementing Fig. 7, Figs. 15 to 18 illustrate how the policy behavior and the privileged-observation distribution change when we manually modify the uncertainty fed into the privileged policy (PP). Consistent with Fig. 7, the PP proves highly sensitive to the uncertainty input: it strictly follows the paradigm of gathering information when the uncertainty is high and exploiting the current privileged observation when the uncertainty is low. Because of this property, PP greatly reduces the behavioral divergence between DP and PP.

## D.4 Sensitivity Studies

To analyze USPL's sensitivity to its hyperparameters, we conduct studies on the newly introduced parameters: those affecting the reward transformation ($C_{\min}$, $C_{\max}$) and the hyperparameter $K$, which controls the magnitude of the perturbation noise applied to $o_p$. We present our sensitivity analysis for these two types of hyperparameters.

First, we analyze the sensitivity to $C_{\min}$ and $C_{\max}$, which jointly define the range of the scaling coefficient applied to the original reward. This coefficient is determined by $\sigma_p$. Specifically, if $\log(\sigma_p) > C_{\min}$, the coefficient reaches its minimum value. As $\sigma_p$ decreases, the coefficient

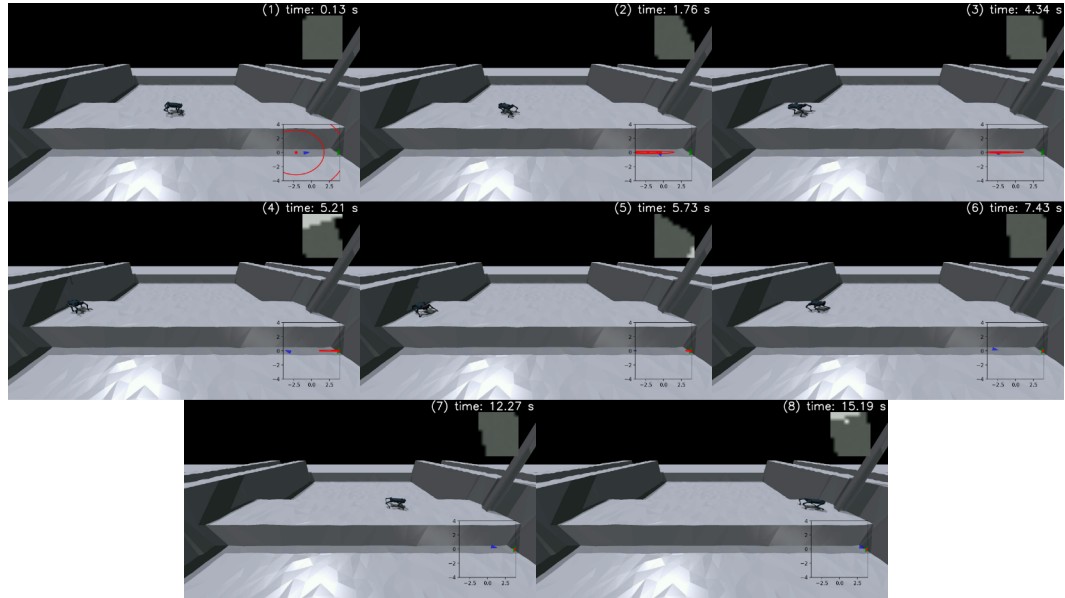

Figure 10: Deployment-policy (DP) behavior and the predicted privileged-observation distribution in *Lateral Choice*. The current time is shown in the top-right corner, and the red ellipse in the bottom-right visualizes the Gaussian distribution (mean and variance) produced by the observation encoder. The star marks the goal, while the triangle marks the robot's position. At the start of the episode (1–2) the predicted distribution collapses into a horizontal line slightly left of center, prompting the robot to explore leftwards (3). Upon reaching the left boundary it realizes the episode has not ended, implying the goal lies to the right; the distribution therefore narrows sharply (4). The robot then moves right and the predicted distribution converges on the goal position (5–8).

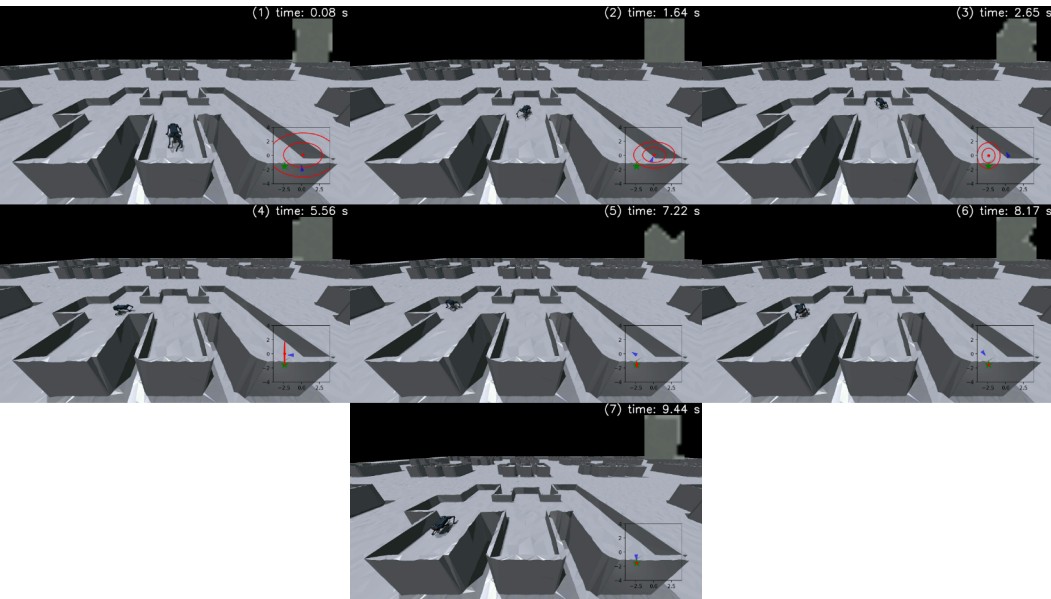

Figure 11: Deployment-policy (DP) behavior and the predicted privileged-observation distribution in *Square Maze*. The layout is as shown in Fig. 10. At the beginning (1–2) the predicted distribution spans a large area. After leaving the start, the robot overshoots the first junction; finding no wall ahead, it deduces that the goal is to its left, and the distribution shifts leftwards (3). The robot turns left (4); at the second junction it again moves forward, discovers no wall (5), and infers the goal is further left. The distribution then collapses onto the goal, and the robot heads straight towards it (5–7).

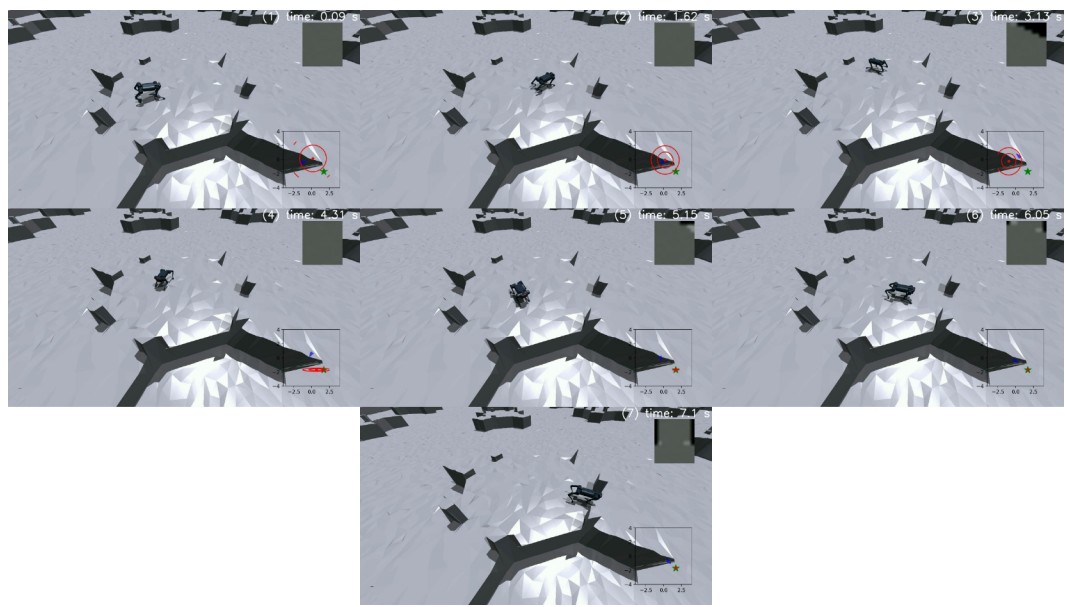

Figure 12: Deployment-policy (DP) behavior and the predicted privileged-observation distribution in *Midpoint Choice*. The layout is as shown in Fig. 10. At the episode start (1–2) the predicted distribution covers nearly the whole arena. The robot circles the central platform (3). Observing that one upper platform lacks a pillar and the other has a very short pillar, it infers that the goal must lie in the lower half (4) and begins exploring downward. After comparing the pillar heights of the two lower platforms, it identifies the lower-right pillar as the second tallest (5) and proceeds directly to the goal (6–7).

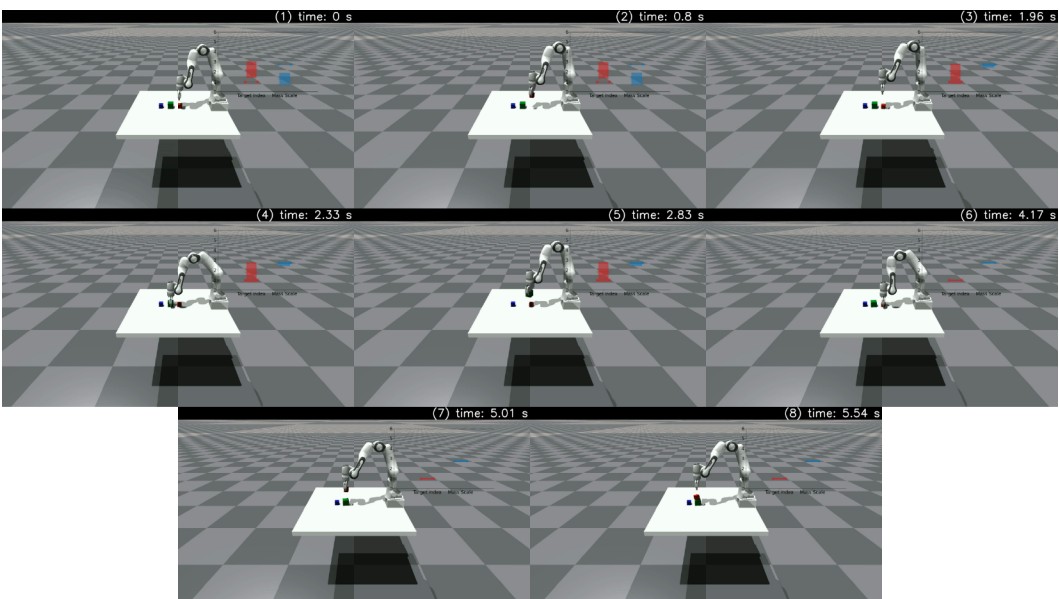

Figure 13: Deployment-policy (DP) behavior and the predicted privileged-observation distribution in *Blind-Mass Stack*. Red and blue bars depict the two-dimensional Gaussian output from the observation encoder; the dashed line between the two triangles indicates the true privileged-observation value. Initially, uncertain which cube is red, the manipulator grasps a red cube (without knowing it is red), records its mass, and puts it down (1–3). To verify its color, it next grasps a green cube with known ID and mass (4–5), uses it to calibrate the force sensor, and, combining this with the earlier measurement, infers that the first cube was red. At this point the red and blue bars converge to the true privileged value (6), and the manipulator successfully stacks the red cube on the green cube.

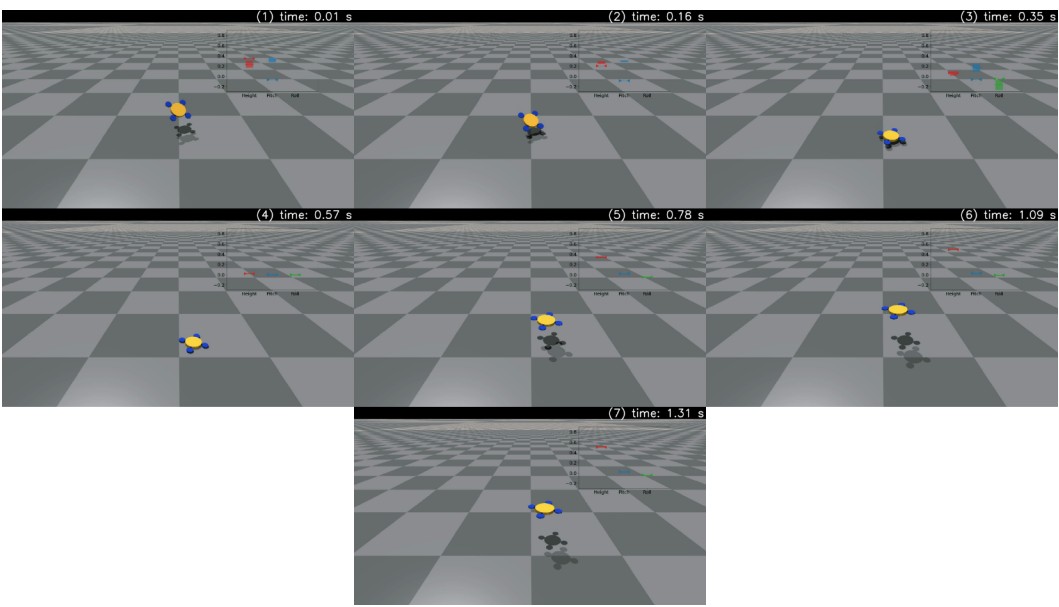

Figure 14: Deployment-policy (DP) behavior and the predicted privileged-observation distribution in *Biased Quadrotor*. Red, blue and green bars represent the observation encoder's predictions of altitude, pitch and roll, respectively; dashed lines between triangles show their true values. At the beginning (1–2) the estimates deviate from the true state, but as the quadrotor descends (3–4) the sensors become calibrated and the predictions align with reality. The quadrotor then takes off again and completes the task (5-7).

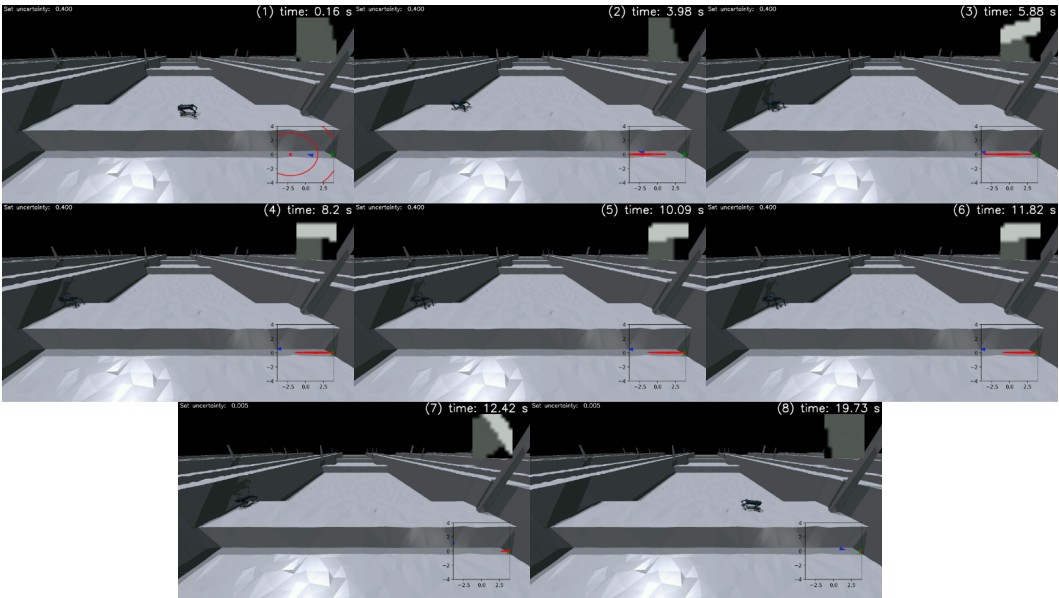

Figure 15: Deployment-policy (DP) behavior and the predicted privileged-observation distribution in *Lateral Choice* with manually adjusted uncertainty. The current time is shown in the top-right corner, while the red ellipse in the bottom-right visualizes the Gaussian output (mean and variance) from the observation encoder. The star marks the goal and the triangle the robot's position; the top-left corner displays the externally supplied uncertainty value. At the start of the episode (1–2) the robot moves left to test whether the goal is there. After reaching the left boundary, it realizes the episode has not ended, so the predicted distribution shifts rightwards (3–4). Because the strategy still perceives high uncertainty, it remains in place and continues exploring (5–6). Only after we set the uncertainty to a lower value does it turn and head directly for the goal (7–8).

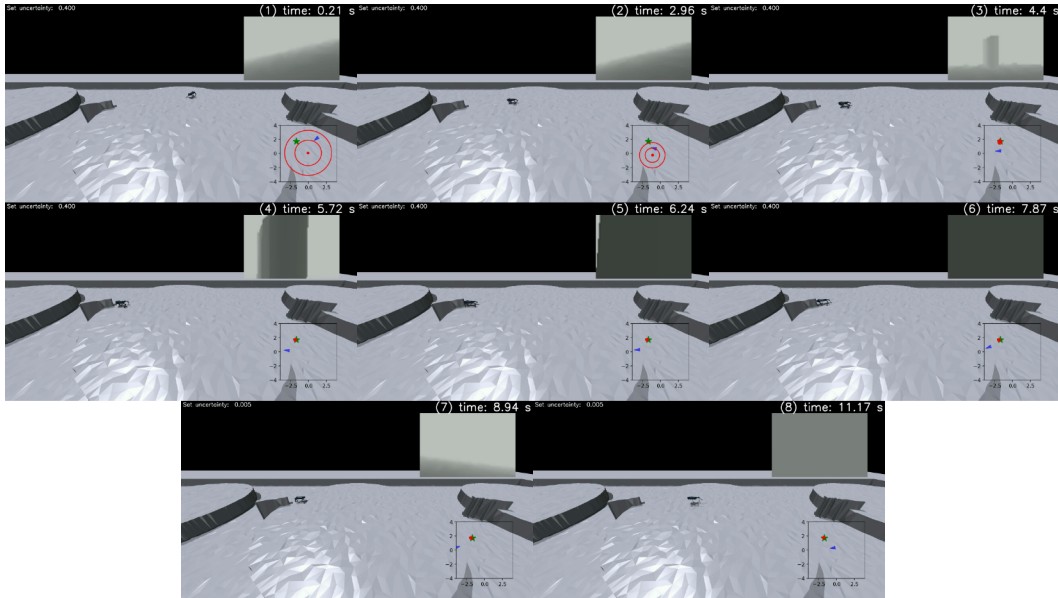

Figure 16: DP behavior and the predicted privileged-observation distribution in *Signpost Nav* under manual uncertainty manipulation. The top-right corner shows time; the bottom-right red ellipse depicts the Gaussian output of the observation encoder. The star indicates the goal, the triangle the robot. The top-left corner shows the current uncertainty input, and the upper-right inset presents the depth-camera view. Early in the episode (1–2) the robot moves toward the signpost to infer the goal's location. Once the signpost appears in the depth image (3) the goal position is immediately inferred, yet the robot continues to approach and then waits near the signpost because the uncertainty is still high (4–6). After we manually lower the uncertainty, it turns back and moves straight toward the goal (7–8).

increases, until $C_{\min} - \log(\sigma_p) > C_{\max}$, at which point the coefficient reaches its maximum value. We tested various combinations of $C_{\min}$ and $C_{\max}$ in both the Signpost Nav and Biased Quadrotor tasks. The results are presented in Tab. 10.

For the Signpost Nav task, the results (first three rows of Tab. 10) show that a narrow range for the scaling coefficient limits the effectiveness of the reward transformation, yielding a success rate below 90%. As the range expands, the success rate consistently remains above 97%. The Biased Quadrotor task exhibits a similar trend: the narrowest range results in a very low success rate (12.5%). However, this task also reveals that an excessively large range can be detrimental. As shown in the last two rows, when the range becomes too wide, performance drops significantly to 14.0% and 53.1%, respectively.

Overall, these results suggest that as long as the range of the scaling coefficient is not excessively large or small, the policy's success rate is largely insensitive to the specific values of $C_{\min}$ and $C_{\max}$. Within an appropriate range, different combinations of these parameters can achieve near-optimal performance.

Table 10: Sensitivity Studies on $C_{\min}$ and $C_{\max}$.

| $C_{\min}$ | $C_{\max}$ | Signpost Nav | Biased Quadrotor |
|---|---|---|---|
| $\log(0.058)$ | $C_{\min} - \log(0.053)$ | $38.6 \pm 2.5$ | $12.5 \pm 2.1$ |
| $\log(0.06)$ | $C_{\min} - \log(0.05)$ | $42.7 \pm 9.3$ | $97.7 \pm 0.2$ |
| $\log(0.075)$ | $C_{\min} - \log(0.04)$ | $87.7 \pm 7.1$ | $97.8 \pm 0.6$ |
| $\log(0.1)$ | $C_{\min} - \log(0.02)$ | $97.6 \pm 0.5$ | $96.0 \pm 0.3$ |
| $\log(0.2)$ | $C_{\min} - \log(0.01)$ | $97.8 \pm 0.2$ | $97.0 \pm 0.4$ |
| $\log(0.4)$ | $C_{\min} - \log(0.005)$ | $98.2 \pm 0.2$ | $14.0 \pm 0.7$ |
| $\log(0.6)$ | $C_{\min} - \log(0.003)$ | $98.1 \pm 0.2$ | $53.1 \pm 31.7$ |

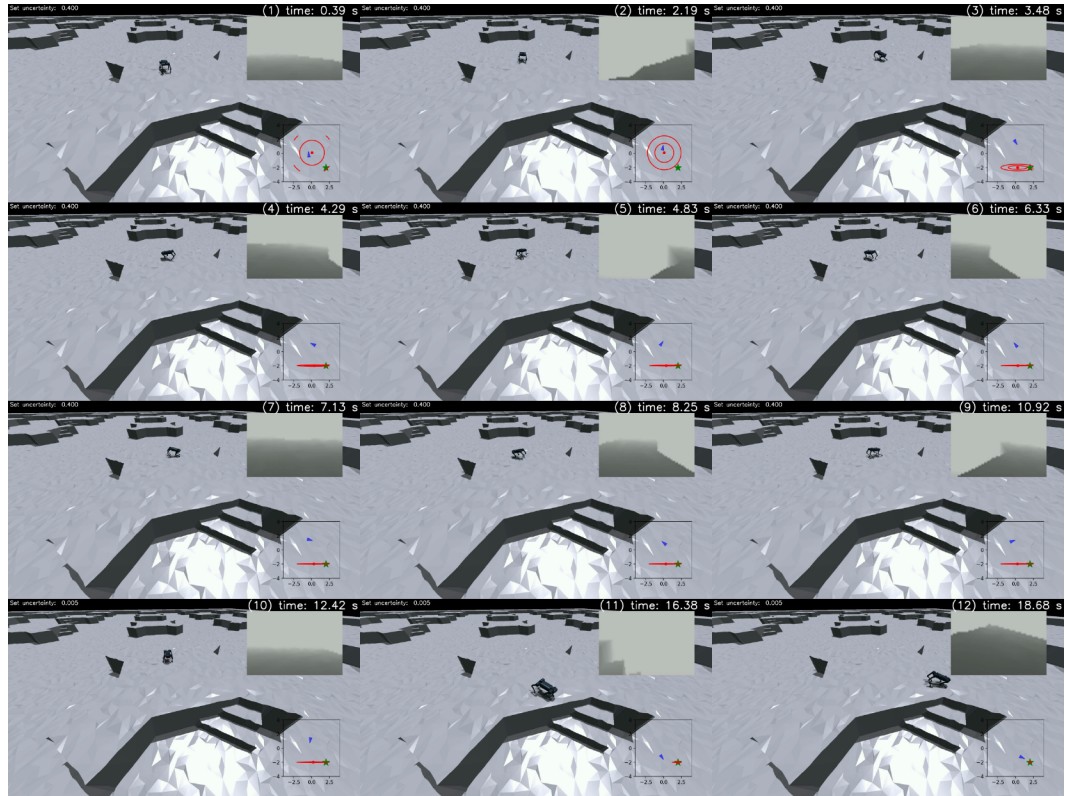

Figure 17: DP behavior and the predicted privileged-observation distribution in *Stairway Search* while manually varying uncertainty. The current time is shown in the top-right, the Gaussian prediction appears as a red ellipse in the bottom-right, the star marks the goal and the triangle the robot. The top-left corner shows the supplied uncertainty value, and the upper-right inset displays the depth-camera image. At the beginning (1–3) the robot approaches the upper platform edge and looks down for stairs. Finding none, the predicted distribution collapses to the lower half. Because the supplied uncertainty remains high, the policy keeps inspecting the lower edge repeatedly (4–9). After we reduce uncertainty at step 10, the robot descends, locates the stairs, and walks to the goal.

Next, we perform a sensitivity analysis on $K$, which controls the magnitude of the perturbation noise applied to $o_p$. The results are presented in Tab. 11. In the Signpost Nav task, the value of $K$ has almost no impact on the policy's success rate. For the Biased Quadrotor task, however, smaller values of $K$ (e.g., $K < 10$) lead to a significant drop in the success rate. For $K \geq 10$, the policy consistently achieves a high success rate. This indicates that USPL performs well across a wide range of $K$ values, and its performance becomes largely insensitive to $K$ once it is sufficiently large ($K \geq 10$).

Table 11: Sensitivity Studies on $K$.

| $K$ | Signpost Nav | Biased Quadrotor |
|---|---|---|
| 1 | $95.7 \pm 1.6$ | $40.7 \pm 20.8$ |
| 2.5 | $98.4 \pm 0.4$ | $9.6 \pm 1.2$ |
| 5.0 | $97.7 \pm 0.5$ | $8.2 \pm 0.5$ |
| 10.0 | $97.8 \pm 0.2$ | $96.7 \pm 0.6$ |
| 20.0 | $98.0 \pm 0.2$ | $97.7 \pm 0.3$ |
| 40.0 | $97.8 \pm 0.3$ | $97.5 \pm 0.3$ |

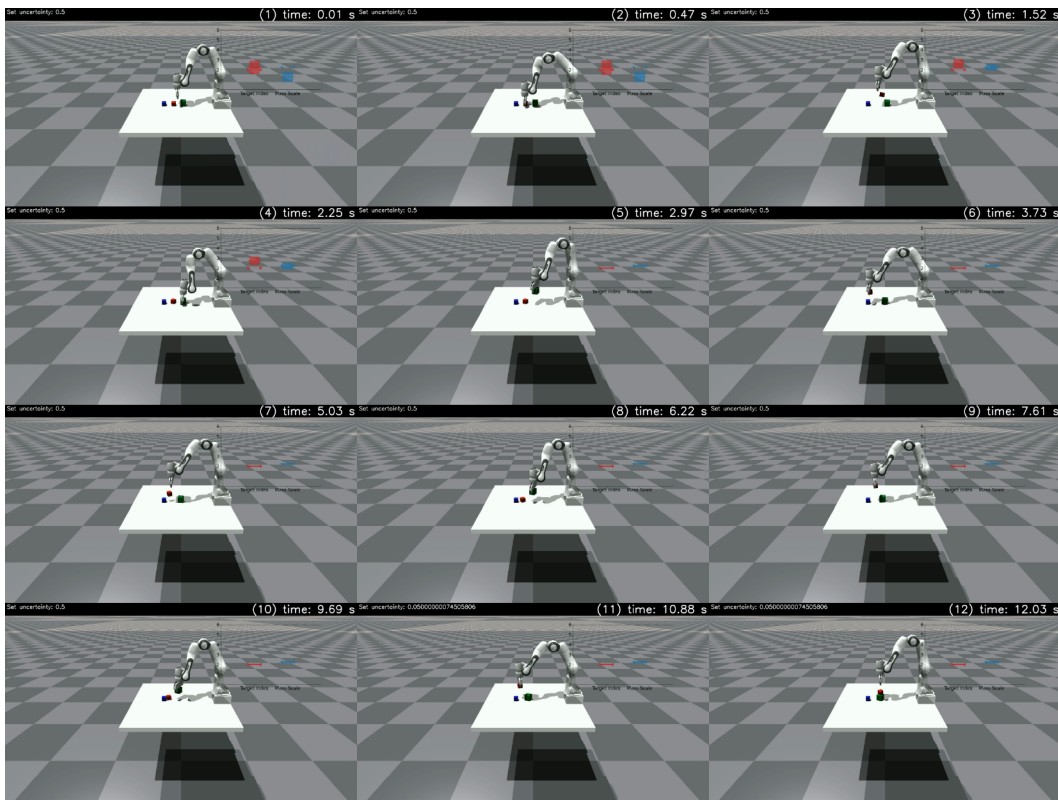

Figure 18: DP behavior and the predicted privileged-observation distribution in *Blind-Mass Stack* with manual uncertainty control. The top-right corner shows time. Red and blue bars represent the two-dimensional Gaussian output of the observation encoder; the dashed line between two triangles indicates the true privileged-observation value. The top-left corner lists the current uncertainty input. To determine which cube is the 0.75 kg red cube, the manipulator sequentially grasps a red and then a green cube (1–5). Although the predicted privileged observation converges to the true value, the strategy still detects high uncertainty and keeps repeating the calibration behavior—grasping red and green cubes alternately (6–10). Only after we lower the uncertainty does it finally pick up the red cube and stack it on the green one.

