# OpenReview forum: "Uncertainty-Sensitive Privileged Learning"
_NeurIPS.cc/2025/Conference — NeurIPS 2025 poster_

### Official Review · Reviewer_dzCL · 2025-06-27

**Clarity:** 3
**Significance:** 3
**Originality:** 3
**Rating:** 4
**Confidence:** 4

**Summary:**

The paper addresses the challenge of behavioural discrepancies between the privileged policy (PP) and the deployment policy (DP) in the context of privileged learning. The authors propose a novel framework, Uncertainty-Sensitive Privileged Learning (USPL), which estimates the prediction uncertainty of privileged observations and then uses prediction uncertainty to guide the PP with reward transformation and privileged observation blurring. Empirical evaluations on nine diverse tasks demonstrate that USPL achieves superior deployment performance, particularly in reducing behavioural discrepancies, when compared to several strong baselines.

**Questions:**

Along with the points mentioned in the section on weakness, I have two further questions.

1. I do not follow the choice of a uniform distribution for the noise in Equation (3). The authors claim that the PP is robust to Gaussian noise, but they do not clarify why uniform noise was chosen instead. If the noise is used to simulate uncertainty or model robustness, Gaussian noise would be the more canonical choice, especially if the observation encoder models uncertainty via Gaussian distributions. Is there any requirement that the noise must be bounded and symmetric for the analysis or experiments? If the uniform noise is meant to be bounded, that rationale should be made explicit. Without such clarification, the design choice feels somewhat arbitrary.

2. In Table 1, the success rate for the Biased Quadrotor task under USPL is reported as nearly perfect with a variance of 0.00, which is suspicious at first glance. Could the authors clarify this point? If the performance is indeed identical across all seeds?

**Ethical Concerns:**

["NO or VERY MINOR ethics concerns only"]

**Final Justification:**

I have no further questions and appreciate the clarifications provided by the authors. My only remaining concern is that the performance of the proposed approach appears to rely on the choice of several hyperparameters. While the authors have included a sensitivity analysis, I still have some reservations about the generalizability of the method. I have increased my score, but I leave the final decision to the AC and other reviewers.

**Limitations:**

See above.

**Quality:**

3

**Strengths And Weaknesses:**

The motivation to reduce behavioural discrepancies between the PP and the DP due to asymmetric observation spaces is clearly stated and, to the best of my knowledge, remains relatively underexplored. The author discusses the limitations of the existing approach (e.g., RL-based fine-tuning DP, regularization) and proposes an original direction of using the prediction uncertainty of privileged observations to guide the PP. I find the idea intuitive and sound. The framework is evaluated across a diverse set of challenging tasks, and the results consistently demonstrate superior performance over the baselines, particularly in reducing behavioural discrepancies.

My major concern lies in the lack of theoretical justification for the proposed framework. It remains unclear to me how the estimated prediction uncertainty directly influences the reduction in behavioural discrepancies between the PP and DP. While the authors provide explanations for the role of uncertainty in guiding the privileged policy, these arguments are largely intuitive. A formal analysis, such as an error bound between uncertainty and discrepancies. For instance, I suspect that the BD could be upper bounded by a function of the uncertainty in the privileged observation prediction, along with other common factors such as the discount factor, the observation noise level, or the dimensionality of the state space. While I agree it could be hard, a further discussion of challenges could be helpful. It would still be valuable for the authors to include a discussion of the key theoretical challenges and possibly sketch out the conditions under which such a bound might hold. This would help clarify the underlying assumptions of the framework and restrictions.

---

> ### Author Rebuttal · Authors · 2025-07-31
>
> ### W1
>
> > Theoretical justification.
>
> We are grateful to the reviewer for this insightful comment regarding the theoretical justification for our proposed framework. Heeding this suggestion, we have developed a formal analysis to address this concern. Specifically, by combining a Lipschitz assumption on the policy with a result from Lemma 4 in [1], we derive the following cumulative error bound that formally connects prediction uncertainty to the behavioral discrepancy between the PP and DP.
>
> Let the policy in USPL be denoted as $\pi(\cdot\mid o_ c, o_ p, \sigma_ p)$, where $o_ c, o_ p,$ and $\sigma_ p$ represent the common observation, privileged observation, and uncertainty estimation, respectively. The key difference between the PP and the DP lies in the source of $o_ p$. Let the privileged observation input to the PP at timestep $h$ be $o_ p^h$ (the ground truth) and the one input to the DP be $\hat{o}_ p^h$ (the prediction). We assume there exists an error bound $\epsilon^h$ such that:
> $$
> \Vert o_ p^h - \hat{o}_ p^h \Vert_ 2 \leq \epsilon^h.
> $$
>
> Here, $\epsilon^h$ represents the upper bound on the prediction error of the privileged observation. We assume that the policy $\pi(\cdot\mid o_ c, o_ p, \sigma_ p)$ is Lipschitz continuous with respect to the privileged observation $o_p$:
>
> $$
> D_ {TV}(\pi(\cdot\mid o_ c, o_ p^h, \sigma_ p), \pi(\cdot\mid o_ c, \hat{o}_ p^h, \sigma_ p)) \leq L(o_ c, \sigma_ p) \Vert o_ p^h - \hat{o}_ p^h \Vert_ 2,
> $$
>
> where $L(o_ c, \sigma_ p)$ is the Lipschitz constant of $\pi$ with respect to $o_ p$ for a given $o_ c$ and $\sigma_ p$. This constant will be large if the policy is highly sensitive to privileged observations. We assume $\epsilon^h$ is known and is used as the uncertainty input to the policy. Let $\pi_ h^{PP} = \pi(\cdot\mid o_c, o_ p^h, \epsilon^h)$ and $\pi_ h^{DP} = \pi(\cdot\mid o_ c, \hat{o}_ p^h, \epsilon^h)$. Under these assumptions, the following inequality holds for the cumulative discrepancy between the state visitation distributions:
>
> $$
> \frac{1}{H}\sum_ {h=1}^H D_ {TV}(d_ h^{\pi^{PP}}, d_ h^{\pi^{DP}}) \leq \sum_ {h=1}^H \mathbb{E}_ {o_ c \sim d_ h^{\pi^{PP}}}\left[L(o_ c, \epsilon^h)\epsilon^h\right],
> $$
>
> where $d_ h^{\pi^{PP}}$ and $d_ h^{\pi^{DP}}$ are the visitation distributions of $o_ c$ at step $h$ for the PP and DP, respectively, and $H$ is the horizon. A detailed derivation of this bound will be provided in the revised paper.
>
> **Relationship between the Error Bound and our USPL Framework:**
>
> This bound formally demonstrates that the cumulative behavioral discrepancy is upper-bounded by a function of the privileged observation prediction error ($\epsilon^h$) and the policy's sensitivity to that information ($L(o_c, \epsilon^h)$). To minimize this compounding error, one can pursue two primary strategies:
>
> * **(S1) Minimize the prediction error $\epsilon^h$**: This involves improving the prediction accuracy of the observation encoder.
> * **(S2) Reduce the policy's sensitivity to the privileged observation**: This involves decreasing the Lipschitz constant $L(o_c, \epsilon^h)$.
>
> In USPL, S1 is directly addressed by the optimization objective of our observation encoder and reward transformation, both of which aim to reduce $\epsilon^h$. S2 is realized through privileged blurring, which adds noise to the privileged observation during training to reduce the policy's over-reliance on it. However, at the beginning of an episode, when the agent's perception of the environment is insufficient, $\epsilon^h$ is typically large. At the same time, solving the task often requires leveraging the privileged observation, which means $L(o_ c, \epsilon^h)$ also needs to be large. To prevent the compounding error from escalating due to both $\epsilon^h$ and $L(o_ c, \epsilon^h)$ being high simultaneously, USPL also achieves an adaptive dependence on privileged observations by taking uncertainty as policy input:
>
> * **When uncertainty ($\approx \epsilon^h$) is high,** the privileged observation is very noisy. Thus, the policy learns to distrust the privileged input (effectively reducing $L(o_ c, \epsilon^h)$). Guided by the transformed reward, it prioritizes exploration to gather information, thereby reducing future uncertainty.
> * **As uncertainty ($\approx \epsilon^h$) decreases,** the privileged observation becomes more reliable. The reward for task completion is magnified by reward transformation. As a result, the policy learns to increase its reliance on the privileged observation (increasing $L(o_c, \epsilon^h)$) to efficiently solve the task.
>
> This coordinated control of both $\epsilon^h$ and $L(o_c, \epsilon^h)$ effectively tightens the upper bound on the cumulative error. This, in turn, minimizes the behavioral discrepancy between the PP and DP, mitigating performance degradation at deployment.
>
> **Empirical Evidence:** As demonstrated by our experiments with manually set uncertainty (Figure 7, Figs. 14-17 in the appendix, and videos in the supplementary material), USPL policy successfully learns this adaptive capability. When provided with high uncertainty, the policy indeed refrains from exploiting the privileged observation—even when the prediction is accurate—and instead prioritizes exploration. It only begins to leverage the privileged observation to complete the task once the uncertainty is reduced, validating our theoretical analysis.
>
> We will add a new subsection in Sec. 3 of the revised paper to elaborate on this theoretical analysis, thereby strengthening the paper's rationale.
>
> [1] Understanding adversarial imitation learning in small sample regime: A stage-coupled analysis. arXiv preprint arXiv:2208.01899.
>
> ### Q1
>
> > The choice of a uniform distribution to randomize the privilieged observations.
>
> We apologize for the confusion. The form of noise perturbation in Eq. (3) is indeed a hyperparameter. We found that when using Gaussian noise ($\mathcal{N}(\cdot\mid 0, \sigma_p^2)$), even with a high fixed uncertainty $\sigma_p$, the PP could sometimes still complete the task without performing information-gathering actions. This led to a significant behavioral discrepancy between the PP and the DP. On the other hand, increasing the noise magnitude using a distribution like $\mathcal{N}(\cdot\mid 0, (K\sigma_p)^2)$ with $K>1$ often resulted in perturbed observations $\hat{o}_p$ that fell outside their practical range and the output range of the observation encoder's mean $\mu_p$. These are values that would never be encountered during deployment, and their frequent occurrence could potentially reduce training efficiency. Moreover, in our experiments, the environment parameters corresponding to $o_p$ are sampled from a bounded uniform distribution. By using uniform noise for perturbation, we can adjust the distribution's bounds to ensure that the final $\hat{o}_p$ remains within a valid range. The $\hat{o}_p$ with the maximum perturbation follows the prior distribution of $o_p$. We will incorporate the aforementioned discussion into Sec. 3.2 of the revised version of the paper.
>
> The table below compares the success rate using the perturbation method from Eq. (3) versus a Gaussian distribution $\mathcal{N}(\cdot\mid 0, (K\sigma_p)^2)$.
>
> | Task             | USPL           | USPL w/ Gaussian Noise |
> | ---------------- | -------------- | ---------------------- |
> | Signpost Nav     | $97.8 \pm 0.2$ | $97.2 \pm 0.3$         |
> | Biased Quadrotor | $96.7 \pm 0.6$ | $17.8 \pm 6.3$         |
>
> In the Signpost Nav task, the performance difference is minor. However, in the Biased Quadrotor task, the performance with Gaussian noise is substantially worse.
>
> ### Q2
>
> > The anomalous success rate standard error for the Biased Quadrotor in Table 1.
>
> Thank you for pointing this issue out. This was indeed a calculation error on our part. The success rates for the 5 seeds were [0.983, 0.955, 0.956, 0.963, 0.977], with a mean of approximately 0.967 and a standard error of 0.0056. We mistakenly forgot to multiply the standard error by 100, which led to the anomalous value in the table. The correct success rate is $96.7 \pm 0.6$. We will correct this in the revised version of our paper.
>
> ---
>
> We are truly grateful for your thoughtful and constructive feedback. Your suggestions were incredibly helpful and have guided us in making significant improvements to the paper. We believe the paper is much stronger now as a result. We hope our responses have thoroughly addressed all your concerns, and we look forward to your feedback. Please do not hesitate to contact us if any points require further discussion.

---

> > ### Comment · Reviewer_dzCL · 2025-08-06
> >
> > Thanks to the author for the clarification. The newly added theoretical guarantee makes it much clearer how the proposed method works. I have no remaining questions.

---

> > > ### Author Response · Authors · 2025-08-06
> > >
> > > Thank you very much for your quick response and positive feedback. We are glad to hear that the added theoretical guarantee helped clarify how our proposed method works. We appreciate your time and valuable input throughout this process.

---

### Official Review · Reviewer_GVJr · 2025-07-02

**Clarity:** 2
**Significance:** 2
**Originality:** 2
**Rating:** 4
**Confidence:** 1

**Summary:**

The paper proposed USPL, using the uncertainty estimation to guide DP to minimize the divergence between DP and PP. Experiments show that USPL can achieve superior performance than baseline methods.

**Questions:**

1. Have you compared the compuation cost across all methods tested, including the baseline methods?
2. If the removing uncertainty estimation does not have the most significant improvement compared with other components as shown in 4.5, is the paper's argument about the importance of uncertainty estimation still valid? Is it possible to apply these components to the baseline methods and also significantly improves the performance?

**Ethical Concerns:**

["NO or VERY MINOR ethics concerns only"]

**Final Justification:**

Thank you for the rebuttal. It solved most concerns and I will keep my score.

**Limitations:**

Yes

**Quality:**

2

**Strengths And Weaknesses:**

Strengths:
1. The motivation is clear and using the uncertainty estimation to contrain the behavior of DP sounds reasonable.
2. The empirical performance of the proposed method is signficantly better than the baseline methods in the environment provided in the paper.
3. Extensive empirical analysis and ablation studies of the proposed method. I am no expert in privileged learning so not sure whether the task range is limited though.

Weaknesses:
1. There seems to be a lack of comparison in terms of the computation cost with the baseline methods, since it includes training and inference of the encoder model.  See Q1.
2. In the ablation studies, the performance improvement seems to largely come from other compoenents like blurring and modifying the reward, which make it questionable that what the main contribution of the propose method should provide. See Q2.

---

> ### Author Rebuttal · Authors · 2025-07-31
>
> ### W1
>
> > Lack of comparison in terms of the computation cost with the baseline methods.
>
> Thank you for your valuable feedback. We agree that a discussion on computational cost is crucial for a complete comparison. In our experiment, all baseline methods were implemented with identical network architectures to ensure a fair comparison. The differences between the baselines lie in the loss functions used for training various modules. We observed that the primary computational overhead stems from environment interaction and network updates, rather than the specific loss calculations. Consequently, the overall computational costs for different methods are quite similar.
>
> We show the average training time per epoch for each method on the Signpost Nav task in the following:
>
> | Method     | Average Training Time per Epoch (s) |
> | :--------- | :---------------------------------- |
> | USPL       | 53.7                                |
> | RMA        | 53.1                                |
> | PrivRecons | 53.2                                |
> | RewReg     | 53.4                                |
> | RESeL-PPO  | 46.8                                |
>
> As shown, RESeL-PPO, which trains the entire DP network end-to-end with a PPO loss, is approximately 13% faster than the other methods that require additional loss computations. The training times for USPL and the other three baselines show no significant differences. We will incorporate this comparison into the revised version of our paper.
>
> ### W2
>
> > In the ablation studies, the performance improvement seems to largely come from other compoenents like blurring and modifying the reward, which make it questionable that what the main contribution of the propose method should provide. Is it possible to apply these components to the baseline methods and also significantly improves the performance?
>
> We sincerely thank the reviewer for this insightful question. You have pinpointed a crucial aspect of our ablation study that merits a more detailed explanation:
>
> 1. We first want to address the question of whether privileged blurring and reward modification could be applied to baseline methods. These components are not standalone techniques, and are **fundamentally reliant on the uncertainty estimation $\sigma_p$**, which is the central contribution of USPL. They cannot be applied to baseline methods that do not estimate this uncertainty. We also recognize, with thanks to your comment, that our original table caption was a source of confusion. The "w/o uncert" variant does, in fact, still compute and use uncertainty for blurring and reward modification. It only omits uncertainty as a direct input to the PP. We will rename this variant to prevent future ambiguity.
>
> 2. Regarding the performance gain comparison of different components, we would like to clarify that our ablation results should be viewed not as a competition between components, but as a demonstration of how they play distinct and complementary roles in a multi-stage process.
>
>    An optimal DP would ideally adapt its behavior online based on its current **Progress of Collecting Privileged Information (PCPI)**, which is inversely correlated with our uncertainty estimation $\sigma_p$. Our framework aims to build such a policy based on PP, and this requires satisfying two essential conditions:
>
>    - First, the PP must **explore and learn the optimal behaviors for different levels of PCPI**. During training, our privileged blurring and reward transformation components work in concert to achieve this foundational goal. **Privileged blurring** simulates states of high uncertainty, forcing the policy to learn how to act when privileged information is unreliable. Concurrently, **reward transformation** incentivizes the policy to actively explore and discover actions that reduce uncertainty (i.e., increase PCPI). The necessity of this exploration stage is evident in Table 2: removing either privileged blurring or reward transformation leads to a collapse in the success rate, proving they are essential for learning the necessary behaviors.
>
>    - Second, the PP must be able to **dynamically select the appropriate behavior online** based on its current PCPI. This is accomplished by providing the uncertainty $\sigma_p$ as a direct input to the policy. The uncertainty serves as a real-time signal, allowing the policy to intelligently switch between information gathering (when uncertainty is high) and task execution (when uncertainty is low). However, this online adaptation is only effective if the optimal behaviors have already been discovered via privileged blurring and reward transformation. This explains the poor performance of `only uncert` in Table 2.
>
>    Incorporating uncertainty as a policy input is crucial for achieving optimality and reliability. To quantify its importance, we analyzed the task failure rate, a key metric for evaluating policy reliability. The results are presented in the table below. As the table shows, even with the powerful exploration capabilities provided by blurring and reward transformation, the policy still has a notable average failure rate of 9.4%. Providing uncertainty as a direct policy input **reduces this failure rate by 80.9%**, bringing it down to just 1.8%.
>
>    This result powerfully illustrates that while blurring and reward shaping are essential for teaching the policy *what to do*, the uncertainty input is the crucial final piece that tells the policy *when to do it*. It is this ability to dynamically adapt based on a direct measure of its own uncertainty that makes the policy reliable, and ultimately more successful.
>
>    | Task                | Failure Rate (w/o Uncert. Input) | Failure Rate (w/ Uncert. Input) | Failure Rate Reduction |
>    | ------------------- | -------------------------------- | ------------------------------- | ---------------------- |
>    | **Signpost Nav**    | 3.0%                             | 2.2%                            | 26.7%                  |
>    | **Square Maze**     | 5.3%                             | 1.5%                            | 71.7%                  |
>    | **Lateral Choice**  | 8.0%                             | 1.2%                            | 85.0%                  |
>    | **Midpoint Choice** | 21.1%                            | 2.3%                            | 89.1%                  |
>    | **Average**         | 9.4%                             | 1.8%                            | 80.9%                  |
>
>    ---
>
>    We are truly grateful for your perceptive feedback. It prompted a deeper analysis that has, we believe, significantly improved the paper. We hope our explanation clarifies the unique role of each component within our framework. We sincerely look forward to any further discussion and hope that these clarifications and new results will lead you to view our contributions even more favorably.

---

> > ### Comment · Reviewer_GVJr · 2025-08-07
> >
> > Thank you for the rebuttal. It solved most concerns and I will keep my score.

---

> > > ### Author Response · Authors · 2025-08-07
> > >
> > > Thank you so much for your reply. we are glad to know our rebuttal helped clarify most of your concerns. We sincerely appreciate all the time and effort you've put into reviewing our work.

---

> ### Author Response · Authors · 2025-08-06
>
> Dear Reviewer GVJr,
>
> Thank you for your thoughtful and constructive review. Your feedback was incredibly helpful and has allowed us to significantly improve our manuscript.
>
> We have incorporated your suggestions and hope our revisions and rebuttal have fully addressed your concerns. We welcome any further discussion and would be happy to clarify any points as the deadline approaches. We hope you find the paper is much improved and would be grateful if you would consider these changes in your updated assessment.
>
> Thank you again for your time and valuable guidance.
>
> Respectfully,
>
> The Authors

---

### Official Review · Reviewer_mnJ9 · 2025-07-02

**Clarity:** 3
**Significance:** 2
**Originality:** 2
**Rating:** 4
**Confidence:** 3

**Summary:**

The authors propose an approach for operating in POMDPs where the belief state (or a privileged information) is inferred via a special encoder. This encoder is trained by predicting belief state (or a privileged information) during training.

**Questions:**

Could the authors more clearly explain how does their work differ from [1] and why it is novel?

**Ethical Concerns:**

["NO or VERY MINOR ethics concerns only"]

**Final Justification:**

The authors addressed some of my comments and have provided an extensive sensitivity analysis of their method.

I think the approach is sound, however, it should be more clearly presented how it differs from the related work as well as provide evidence of its scalability to high-dimensional situations.

**Limitations:**

Explained above.

**Paper Formatting Concerns:**

No concerns.

**Quality:**

3

**Strengths And Weaknesses:**

The proposed approach bears strong similarity to a method described in [1]. In [1], similarly to the current work, the authors have privileged information during training (belief state) and they train a special network (belief network) to predict the belief state (in the terminology of this work, it is called privileged information). In [1], there is also a partial information which can be available during sampling (called a "cue"), which is called "target information" in the current work. The difference between [1] and current work is that in [1], the authors use RNNs, while here the setting is not limited to RNNs and there is an additional blurring of the privileged information.

In light of [1], the novelty of the current work is quite limited, and the authors should discuss the novelty more explicitly in their paper.

The experiments of the current work are promising but the results are still limited to low-dimensional privileged observations (but [1] would also suffer from the same limitations).

References:

[1] Meta reinforcement learning as task inference, Jan Humplik, Alexandre Galashov, Leonard Hasenclever, Pedro A. Ortega, Yee Whye Teh, Nicolas Heess, 2019

---

> ### Author Rebuttal · Authors · 2025-07-31
>
> ### W1
>
> > Could the authors more clearly explain how does their work differ from [1] and why it is novel?
> >
> > [1] Meta reinforcement learning as task inference, Jan Humplik, Alexandre Galashov, Leonard Hasenclever, Pedro A. Ortega, Yee Whye Teh, Nicolas Heess, 2019
>
>
> We thank the reviewer for pointing out this relevant work. USPL is fundamentally distinct from [1] as USPL is a privileged learning (PL) framework, while [1] is not. This core difference leads to distinct training methodologies, primary challenges, and novel contributions. We highlight three key distinctions:
>
> 1. Usage of privileged observations
>
> - **USPL:** During training, the Privileged Policy (PP) has **direct, real-time access** to privileged observations. This allows the PP to focus on completing the task without needing to learn active information-gathering behaviors, leading to a more efficient RL training process. Our primary focus is then on bridging the "behavioral divergence" between this PP and a deployment policy that operates on target observations.
> - **Method in [1]:** The policy **never** directly uses privileged information as an input. Instead, privileged data is used solely to supervise an auxiliary loss for a belief encoder. Consequently, their single policy must learn complex, active information-gathering behaviors to infer the missing information, making the policy training significantly more challenging.
>
> 2. Core technical contribution
>
> - **USPL:** A central contribution of our work is the **prediction and utilization of uncertainty**. We explicitly model the uncertainty of the privileged observation prediction and leverage this uncertainty to mitigate behavioral divergence.
> - **Method in [1]:** The concept of prediction uncertainty is **entirely absent**. Their work focuses on task inference through belief state representation without considering the reliability of that belief.
>
> 3. Empirical comparison
>
> - Our baseline "PrivRecons" in Table 1 is conceptually analogous to the approach in [1]. PrivRecons uses a observation encoder trained via supervised learning to predict privileged observations, and a single policy uses this encoder's output for both training and deployment.
>
> - The experimental results clearly show that USPL substantially outperforms PrivRecons. This result empirically validates the benefits of our privileged learning framework and our novel use of uncertainty, highlighting the advantages over the methodology presented in [1].
>
> We hope this comparison clarifies the distinctions of our approach and its contributions relative to [1]. We will include the comparisons with [1] in the revised version of our paper.
>
> ### W2
>
> > The experiments are still limited to low-dimensional privileged observations.
>
> This is indeed a known limitation of the current work. Extending USPL to scenarios with high-dimensional, redundant, and time-varying privileged observations is indeed a challenging and important direction for future research. Without additional techniques, the performance of USPL in such settings would heavily depend on the capacity and learning efficiency of the observation encoder.
>
> A promising approach to address this would be to integrate dimensionality reduction techniques. For instance, [2] and [3] introduced a privileged encoder that maps high-dimensional privileged observations into a low-dimensional latent space, and they have demonstrated excellent results. Similarly, USPL could adopt such a method to first project the privileged observations into a compact latent space and then train the observation encoder to predict this lower-dimensional representation.
>
> We will continue to conduct more in-depth research on this limitation in our future work. We will also extend the discussion of this limitation in the revised version of our paper.
>
>
>
> [2] RMA: Rapid motor adaptation for legged robots. RSS 2021.
>
> [3] Deep whole-body control: Learning a unified policy for manipulation and locomotion. CoRL 2022.
>
> ---
>
> We would like to express our sincere gratitude once again for your insightful and constructive suggestions. We hope that our response and the proposed revisions have thoroughly addressed your concerns. We believe the paper is significantly stronger as a result of your guidance, and we would be grateful if you would consider these clarifications in your final assessment. Please do not hesitate to let us know if any further questions arise.

---

> ### Author Response · Authors · 2025-08-06
>
> Dear Reviewer mnJ9,
>
> We would like to extend our sincerest gratitude for your incredibly insightful and thoughtful review. Your constructive comments have been invaluable, providing us with a clear path to significantly strengthen our manuscript.
>
> We hope that our rebuttal and planned revisions have thoroughly addressed your concerns. As the discussion period is drawing to a close, we wanted to reach out one more time. We remain fully available and would be genuinely delighted to provide any further clarification you might need.
>
> If our responses and revisions have successfully resolved your initial concerns, we would be deeply grateful if you would consider these improvements in your final evaluation of our work.
>
> Thank you once again for your invaluable guidance and engagement in this process.
>
> Respectfully,
>
> The Authors

---

> > ### Comment · Reviewer_mnJ9 · 2025-08-07
> > **Response**
> >
> > Dear authors, thank you for your reply. In light of your response, I'm increasing my score.

---

> > > ### Author Response · Authors · 2025-08-07
> > >
> > > Thank you for your positive feedback and for reconsidering our paper. We are very grateful for your time and constructive comments.

---

### Official Review · Reviewer_ru4E · 2025-07-03

**Clarity:** 2
**Significance:** 3
**Originality:** 2
**Rating:** 4
**Confidence:** 3

**Summary:**

This paper considers the problem of learning an effective deployment policy from partial observations that exhibit less behavioral divergence compared to the privileged policy, which has access to privileged observations during training. This work proposes to incorporate the uncertainty of the (partial) observation encoder (that predicts the privileged observation) into the privileged policy learning. The idea is that high uncertainty will drive the policy to collect more information that would reduce the behavioral discrepancy. Experimental results on different environments show the effectiveness of the method.

**Questions:**

The limitation section talks about the elimination of the DP encoder during PP training. Even if you opt for a proxy encoder, does it not need to be trained alongside PP?

There is much room for improvement in the writing part. Sections 1 and 2 often sound repetitive.

**Ethical Concerns:**

["NO or VERY MINOR ethics concerns only"]

**Final Justification:**

This paper incirporates uncertainty estimation within the privileged learning farmework in a simple and effective way. The addition of non-robotic experiments and hyperparameter sensitivity analysis added during the rebuttal provide more insights. I believe adopting a more theoretical discussion along with additional experiments such as other means of corrupting privileged information would strengthen the paper.

**Limitations:**

Yes.

**Quality:**

3

**Strengths And Weaknesses:**

### Strengths:
1. The idea of injecting uncertainty into the privileged learning is interesting. This enables to capture the discrepancy of the deployment scenarios and facilitates generalization. Also, the controlled blurring of privileged information helps to imitate a deployment-level scenario.

2. The additional loss functions to learn the observational encoder are simple and easy to implement.

3. The experiments cover image and non-image observations, and the evaluations show that the deployment policy can achieve performance similar to the privileged policy. Also, the uncertainty estimation analysis and ablation study sections provide meaningful and rational insights.


### Weaknesses:

1. Eqs 3 and 4 introduce new hyperparameters, thus making it more critical to optimize. I do not see any sensitivity analysis to those hyperparameters.

2. The experiments are limited to robotic environments. It would be nice to see some experiments on other applications of privileged learning.

---

> ### Author Rebuttal · Authors · 2025-07-31
>
> ### W1
>
> > Lack of sensitivity analysis to hyperparameters.
>
> Thank you for raising this important point. Our main hyperparameters include those that affect the reward transformation ($C_{\min}$, $C_{\max}$) and the hyperparameter $K$, which controls the magnitude of the perturbation noise to $o_p$. We present the results of our sensitivity analysis for these two types of hyperparameters.
>
> First, we conduct an sensitivity study on $C_{\min}$ and $C_{\max}$, which affect the scale coefficient for the original reward. If $\log(\sigma_p) > C_{\min}$, the scale coefficient reaches its minimum. Conversely, the scale coefficient increases as $\sigma_p$ decreases, until $C_{\min} - \log(\sigma_p) > C_{\max}$, at which point the coefficient reaches its maximum and stops increasing. Therefore, $C_{\min}$ and $C_{\max}$ jointly determine the range of the scale coefficient. We tested different combinations of $C_{\min}$ and $C_{\max}$ in both the Signpost Nav and Biased Quadrotor tasks, with the results shown in the table below.
>
> For the Signpost Nav task, the first three rows show that a relatively small range for the scale coefficient limits the impact of the reward transformation, resulting in a task success rate below 90%. As the range expands, the success rate consistently stays above 97%. The Biased Quadrotor task exhibits a similar trend at the lower end of the range, with the narrowest range resulting in a very low success rate ($12.5\%$). However, it also reveals that an overly large range can be detrimental. As shown in the last two rows, when the range becomes too wide, the performance drops significantly to $14.0\%$ and $53.1\%$, respectively.
>
> Overall, the results suggest that as long as the range of the scale coefficient is not excessively large or small, the policy's success rate is largely insensitive to the specific values of $C_{\min}$ and $C_{\max}$. Within an appropriate range, different combinations of these parameters are capable of achieving near-optimal performance.
>
> | $C_{\min}$    | $C_{\max}$             | Signpost Nav   | Biased Quadrotor |
> | ------------- | ---------------------- | -------------- | ---------------- |
> | $\log⁡(0.058)$ | $C_{\min}-\log(0.053)$ | $38.6 \pm 2.5$ | $12.5\pm 2.1$    |
> | $\log(0.06)$  | $C_{\min}-\log(0.05)$  | $42.7\pm 9.3$  | $97.7\pm 0.2$    |
> | $\log(0.075)$ | $C_{\min}-\log(0.04)$  | $87.7\pm 7.1$  | $97.8\pm 0.6$    |
> | $\log(0.1)$   | $C_{\min}-\log(0.02)$  | $97.6\pm 0.5$  | $96.0\pm 0.3$    |
> | $\log(0.2)$   | $C_{\min}-\log(0.01)$  | $97.8 \pm 0.2$ | $97.0\pm 0.4$    |
> | $\log(0.4)$   | $C_{\min}-\log(0.005)$ | $98.2\pm 0.2$  | $14.0\pm 0.7$    |
> | $\log(0.6)$   | $C_{\min}-\log(0.003)$ | $98.1\pm 0.2$  | $53.1\pm 31.7$   |
>
> Next, we perform a sensitivity analysis on $K$, which affects the magnitude of the perturbation noise to $o_p$. The results are in the table below. The results in Signpost Nav show that the value of $K$ had almost no impact on the policy's success rate. As for the Biased Quadrotor task, the results show that a smaller value of $K$ leads to a significant drop in the success rate, whereas for $K \geq 10$, the policy consistently achieves a high success rate. USPL performs well across a wide range of values for $K$. After $K \geq 10$, the policy's success rate is largely insensitive to the value of $K$.
>
> | $K$  | Signpost Nav   | Biased Quadrotor |
> | ---- | -------------- | ---------------- |
> | 1    | $95.7 \pm 1.6$ | $40.7\pm 20.8$   |
> | 2.5  | $98.4 \pm 0.4$ | $9.6\pm 1.2$     |
> | 5.0  | $97.7 \pm 0.5$ | $8.2\pm 0.5$     |
> | 10.0 | $97.8 \pm 0.2$ | $96.7 \pm 0.6$   |
> | 20.0 | $98.0 \pm 0.2$ | $97.7\pm 0.3$    |
> | 40.0 | $97.8 \pm 0.3$ | $97.5\pm 0.3$    |
>
> We will include these results and the corresponding discussion in the revised version of our paper.
>
> ### W2
>
> > It would be nice to see some experiments on other applications of privileged learning.
>
> Thank you for this constructive suggestion. We agree that demonstrating the applicability of our method beyond robotics environments would significantly strengthen the paper. As a result, we test our method on the challenging credit assignment task discussed in [1], Active T-Maze (with the credit assignment length set to 500). Active T-Maze is a grid-world task where the agent's navigable area is shaped like a "T" rotated 90 degrees clockwise (as in Fig. 1 of [1]). The goal is located at either the top-right or bottom-right corner. The optimal policy for the agent is to first move one step left to an "oracle" position to reveal the goal's location, then move right to a junction, and finally decide whether to move up or down based on the privileged information it received. [1] found that traditional Transformer-based and RNN-based RL methods struggle to solve such credit assignment tasks efficiently. We approach the Active T-Maze task from a privileged learning perspective, with the results shown below. In this experiment, USPL was trained with 5 different random seeds, while other baselines were trained with 3 random seeds due to limited computational resources.
>
> | Method       | PrivRecons    | RESeL-PPO     | RewReg        | RMA           | USPL (ours)    | RMA-p           | USPL-p         |
> | ------------ | ------------- | ------------- | ------------- | ------------- | -------------- | --------------- | -------------- |
> | Success Rate | $50.7\pm 0.1$ | $43.4\pm 2.1$ | $49.5\pm 0.6$ | $51.4\pm 0.2$ | $100.0\pm 0.0$ | $100.0 \pm 0.0$ | $100.0\pm 0.0$ |
>
> As the results show, USPL achieves a perfect success rate on this highly challenging task, while the other baselines only achieve around a 50% success rate. We hypothesize that USPL's explicit uncertainty input and reward transformation significantly alleviate the difficulty of credit assignment. These results demonstrate the effectiveness of USPL on credit assignment problems. We will add these findings and discussion to the revised version of our paper.
>
> [1] When Do Transformers Shine in RL? Decoupling Memory from Credit Assignment. NeurIPS 2023.
>
> ### Q1
>
> > Even if you opt for a proxy encoder, does it not need to be trained alongside PP?
>
> You are right, the proxy encoder does need to be trained alongside the PP. However, this proxy encoder would be more lightweight than the observation encoder we currently use, leading to faster training. This is especially true for image-based inputs. Currently, training the PP requires processing images with the observation encoder, which is slow. By contrast, using a proxy encoder to process the generally lower-dimensional historical privileged observations would be much more efficient.
>
> ### Q2
>
> > There is much room for improvement in the writing part. Sections 1 and 2 often sound repetitive.
>
> Thank you for this valuable feedback. We will thoroughly revise the writing of the entire paper in the next version, with a particular focus on Sections 1 and 2, to improve clarity and readability.
>
> ---
>
> Thank you once again for your detailed review and constructive suggestions. Your feedback has been crucial in helping us enhance the quality of our paper. In light of these clarifications and additional experiments, we would be grateful if you would consider a higher rating. We would be more than happy to provide any further clarifications if needed.

---

> > ### Comment · Reviewer_ru4E · 2025-08-07
> >
> > Thanks to the authors for addressing my concerns and adding results to assess sensitivity to the hyperparameters. Based on that, I would like to keep my scores.

---

> > > ### Author Response · Authors · 2025-08-08
> > >
> > > Thank you for your reply. We are grateful for your time and insightful feedback on our paper!

---

> ### Author Response · Authors · 2025-08-06
>
> Dear Reviewer ru4E,
>
> We would like to express our profound appreciation for your perceptive and thorough review. Your suggestions were truly instrumental, offering critical insights that have allowed us to substantially enhance the quality and clarity of our manuscript.
>
> We hope that our rebuttal and the accompanying revisions have fully addressed your concerns. Mindful that the discussion window is concluding soon, we wanted to make ourselves available for any final questions you might have. We would be more than happy to engage further on any point. It is our sincere hope that these revisions, which were directly motivated by your feedback, have strengthened the paper to your satisfaction. We would be very grateful if you would take these developments into account in your updated assessment.
>
> Thank you once more for your expert guidance and for your constructive partnership in refining this work.
>
> Respectfully,
>
> The Authors

---

### Author Response · Authors · 2025-08-05
**General Response**

Dear Reviewers,

We would like to extend our sincerest gratitude to all reviewers for your time, effort, and invaluable feedback on our manuscript. We are very encouraged that you found merit in our work and deeply appreciate the constructive criticism, which has provided us with a clear path for significant improvement.

We are delighted to hear the encouraging feedback from the reviewers:

- The motivation is clear, well-presented, and relatively underexplored (`GVJr, dzCL`).
- The core idea is sound, interesting, and easy to implement (`dzCL, ru4E`).
- The proposed method demonstrates significant advantages in extensive and challenging tasks (`GVJr, dzCL`).
- The experimental results provide meaningful and rational insights (`ru4E`).

At the same time, we recognize the areas where our paper can be strengthened, as you have pointed out:

- The relationship with meta-RL methods using privileged observations was not sufficiently clarified (`mnJ9`).
- The paper lacks theoretical analysis (`dzCL`).
- The experimental section could be enhanced with hyperparameter sensitivity analysis, non-robotic environment results, and a comparison of computational costs (`ru4E, GVJr`).
- Certain sections suffered from a lack of clarity or typos (`All Reviewers`).

Based on your thoughtful suggestions, we plan to make the following revisions to our paper to address these points comprehensively.

1. Clarifying relationship to prior work (`mnJ9`):

We will revise Section 2.2 (Related Work) to include a detailed comparison with meta-RL methods that leverage privileged observations, especially [1].

2. Addition of theoretical analysis (`dzCL`):

We will add a new subsection in Section 3 to introduce a theoretical analysis. Specifically, we will present the cumulative error bound that formally connects privileged observation prediction error to the behavioral discrepancy between the PP and DP.

3. Expanding experimental evaluation:

We will enhance the experimental section with the following additions:

- **Hyperparameter Sensitivity Analysis** (`ru4E`): We will include a sensitivity analysis for key hyperparameters such as Cmin, Cmax, and K in environments like Signpost Nav and Biased Quadrotor.
- **Non-Robotic Environment** (`ru4E`): We will add new experimental results in a non-robotic setting, **Active T-Maze**, to demonstrate the broader applicability of USPL.
- **Computational Cost Comparison** (`GVJr`): We will provide a comparative analysis of the **training time overhead** for USPL against all baselines.

4. Improving clarity and presentation:

We are committed to improving the overall quality of the manuscript.

- **Overall Writing** (`ru4E`): We will thoroughly revise the writing of the entire paper, with a particular focus on **Sections 1 and 2**, to improve clarity and readability.
- **Expanding Discussions in Limitations** (`ru4E, mnJ9`): In the Limitation part, we will add further explanation regarding the purpose of eliminating the DP encoder during PP training. We will also expand the discussion on the limitations related to high-dimensional privileged observations.
- **Clarification of Ablation Study** (`GVJr`): We will rename the ablation component `uncert` to `uncert input` for better clarity and will expand our analysis of the ablation results to offer deeper insights.
- **Justification for Noise Distribution** (`dzCL`): We will provide a clearer justification for our choice of a uniform distribution for the noise in Equation (3). Furthermore, we will add experimental results comparing the performance of uniform versus Gaussian noise to the revised paper.
- **Typos** (`dzCL`): We will correct the anomalous standard error value reported for the Biased Quadrotor environment in Table 1.

We are confident these revisions have substantially strengthened the manuscript, and we are sincerely grateful for the insightful comments that made this improvement possible.

**Given that the discussion period will conclude shortly, we would be grateful for the opportunity to address any final questions you may have. Please let us know if anything requires clarification, as we are eager to continue the discussion.**

Thank you once again for your time and expertise.

Best regards,

The Authors



[1] Jan Humplik, Alexandre Galashov, Leonard Hasenclever, Pedro A. Ortega, Yee Whye Teh, Nicolas Heess. Meta reinforcement learning as task inference. 2019.

---

### Decision · Program_Chairs · 2025-09-17

**Decision:**

Accept (poster)

**Comment:**

In this paper, the authors quantify the deployement policy’s information-gathering progress by estimating the prediction uncertainty of privileged observations reconstructed from partial observations, and propose the framework of Uncertainty-Sensitive Privileged Learning (USPL). This framework feeds this uncertainty estimation to the privileged policy (PP) and combines reward transformation with privileged-observation blurring, driving the PP to choose actions that actively reduce uncertainty and thus gather the necessary information. Experiments across nine tasks demonstrate that USPL significantly reduces the behavioral discrepancies, achieving superior deployment performance compared to baselines. The reviewers have stated that the idea of injecting uncertainty into priviledge learning is interesting. The additional loss functions used to learn the observational encoder are simple and easy to implement. Moreover, the experiments cover image and non-image observations, and show that the deployment policy can achieve performances similar to those of the privileged policy. Also, the uncertainty estimation analysis and ablation study sections provide meaningful and rational insights. Furthermore, the motivation is clear and the empirical performance of the proposed method is signficantly better than the baseline methods in the environment provided in the paper. Specifically, the results consistently demonstrate superior performance over the baselines. Overall I believe that this is a nice paper that will get the attention of the community working on learning under privileged information and RL.